# Mutant huntingtin impairs PNKP and ATXN3, disrupting DNA repair and transcription

Rui Gao[1], Anirban Chakraborty[2], Charlene Geater[3], Subrata Pradhan[1], Kara L Gordon[4], Jeffrey Snowden[1], Subo Yuan[5], Audrey S Dickey[4], Sanjeev Choudhary[6], Tetsuo Ashizawa[7], Lisa M Ellerby[8], Albert R La Spada[4], Leslie M Thompson[3,9], Tapas K Hazra[2], Partha S Sarkar[1,5]*

[1]Department of Neurology, University of Texas Medical Branch, Galveston, United States; [2]Department of Internal Medicine, University of Texas Medical Branch, Galveston, United States; [3]Department of Psychiatry and Human Behavior and the Sue and Bill Gross Stem Cell Center, University of California, Irvine, Irvine, United States; [4]Department of Neurology, Duke University School of Medicine, Durham, United States; [5]Department of Neuroscience, Cell Biology and Anatomy, University of Texas Medical Branch, Galveston, United States; [6]Department of Biochemistry, Cell Biology and Genetics, Sam Houston State University, Huntsville, United States; [7]Department of Neurology, Houston Methodist Research Institute, Houston, United States; [8]Buck Institute for Research on Aging, Novato, United States; [9]Department of Neurobiology and Behavior, University of California, Irvine, Institute for Memory Impairments and Neurological Disorders, Irvine, United States

*For correspondence:
pssarkar@utmb.edu

**Competing interests:** The authors declare that no competing interests exist.

**Abstract** How huntingtin (HTT) triggers neurotoxicity in Huntington's disease (HD) remains unclear. We report that HTT forms a transcription-coupled DNA repair (TCR) complex with RNA polymerase II subunit A (POLR2A), ataxin-3, the DNA repair enzyme polynucleotide-kinase-3'-phosphatase (PNKP), and cyclic AMP-response element-binding (CREB) protein (CBP). This complex senses and facilitates DNA damage repair during transcriptional elongation, but its functional integrity is impaired by mutant HTT. Abrogated PNKP activity results in persistent DNA break accumulation, preferentially in actively transcribed genes, and aberrant activation of DNA damage-response ataxia telangiectasia-mutated (ATM) signaling in HD transgenic mouse and cell models. A concomitant decrease in Ataxin-3 activity facilitates CBP ubiquitination and degradation, adversely impacting transcription and DNA repair. Increasing PNKP activity in mutant cells improves genome integrity and cell survival. These findings suggest a potential molecular mechanism of how mutant HTT activates DNA damage-response pro-degenerative pathways and impairs transcription, triggering neurotoxicity and functional decline in HD.
DOI: https://doi.org/10.7554/eLife.42988.001

## Introduction

Huntington's disease (HD) is an autosomal dominant neurodegenerative disorder caused by a CAG triplet repeat expansion in exon 1 of the *HTT* gene that is translated into polyglutamine (polyQ) sequences in the huntingtin (HTT) protein which leads to progressive deterioration of cognitive and motor functions (The Huntington's Disease [*MACDONALD, 1993*; *Ross and Tabrizi, 2011*; *Vonsattel and DiFiglia, 1998*]). The polyQ expansion in the mHTT protein leads to progressive degeneration most overly affecting γ-aminobutyric acid (GABA)-releasing striatal neurons and

**eLife digest** Our DNA encodes the instructions to make proteins, which then go on to perform many crucial roles in the cell. Breakages and damage to DNA occur over time, and if uncorrected, they can make the instructions illegible or incorrect. A build-up of damages can be harmful – for example, DNA damage from excessive UV light exposure can cause skin cancer. Luckily, cells contain DNA repair complexes, protein machines that surveil DNA and correct errors or breakages.

An accumulation of DNA breakages is thought to contribute to the development of Huntington's disease, a devastating and currently incurable condition where brain cells slowly die. The immediate cause of Huntington's disease is well known: Huntington's patients have an abnormal, mutant version of a protein called huntingtin. However, it is still unclear how the mutant huntingtin causes the symptoms of the disease and participates in cell death.

Gao et al. carefully studied the proteins that huntingtin physically interacts with. The experiments revealed that huntingtin is part of a newly identified DNA repair complex that fixes breakages in DNA as the molecule is 'read' by the cell. The presence of the normal huntingtin protein promoted DNA repair. However, when the healthy huntingtin was replaced with the mutant version found in Huntington's disease, the activity of the DNA repair complex was greatly reduced. This resulted in a build-up of DNA errors, triggering a series of events that ultimately led to cell death. In addition, in mice engineered to produce the mutant version of huntingtin, the accumulation of DNA damage was particularly important in two brain regions that are severely damaged in patients with Huntington's disease.

There is currently no effective treatment for Huntington's disease. However, understanding how the mutant huntingtin damages brain cells may provide new targets for future therapies. More broadly, several other brain disorders share similarities with Huntington's disease, and it remains to be seen whether the same mechanisms could be at work in all these conditions.

DOI: https://doi.org/10.7554/eLife.42988.002

glutamatergic cortical neurons, although neuronal dysfunction and tissue atrophy in other brain regions is also present (*Vonsattel and DiFiglia, 1998*; *Ross and Tabrizi, 2011*). Altered conformation of the mutant protein is reported to reduce normal function of the protein as well as facilitate aberrant protein-protein interactions or subcellular localization, leading to neurotoxicity. Among the numerous molecular interactions and signaling pathways implicated in HD pathomechanism, transcriptional dysregulation (*Jimenez-Sanchez et al., 2017*; *Ross and Tabrizi, 2011*; *Valor, 2015*), mitochondrial (mt) dysfunction (*Shirendeb et al., 2011*; *Siddiqui et al., 2012*), DNA strand break accumulation, and atypical ataxia telangiectasia-mutated (ATM) pathway activation, involved in the DNA damage response (*Bertoni et al., 2011*; *Giuliano et al., 2003*; *Illuzzi et al., 2009*; *Xh et al., 2014*), have emerged as key players in HD-related neuronal dysfunction. Genetic or pharmacological ablation of ATM activity to ameliorate the consequence of aberrant ATM activation decreased neurotoxicity in HD animal models and HD induced pluripotent stem cells, respectively (*Xh et al., 2014*), supporting the emerging view that inappropriate and chronic DNA damage-response (DDR) pathway activation is a critical contributor to HD pathogenesis. Although, recent genome-wide association (GWA) studies and genetic data from other sources suggest that DNA damage and repair pathways are central to the pathogenesis of HD and other diseases associated with CAG repeat expansion (*Bettencourt et al., 2016*; *Lee et al., 2015*), the perplexing questions that remain to be elucidated include how polyQ expansion induces DNA strand breaks, activates the DDR pathway, and disrupts transcription. It is also unclear whether transcriptional dysregulation and atypical ATM activation are mechanistically interconnected. We recently reported that the wild-type (wt) form of the deubiquitinating enzyme ataxin-3 (wtATXN3) enhances the activity of polynucleotide kinase-3'-phosphatase (PNKP), a bifunctional DNA repair enzyme with both 3'-phosphatase and 5'-kinase activities that processes unligatable DNA ends to maintain genome integrity and promote neuronal survival. In contrast, mutant ATXN3 (mATXN3) abrogates PNKP activity to induce DNA strand breaks and activate the DDR-ATM→p53 pathway, as observed in spinocerebellar ataxia 3 (SCA3; *Chatterjee et al., 2015*; *Gao et al., 2015*). Furthermore, we recently reported that PNKP plays a

key role in transcription-coupled base excision repair (TC-BER) and transcription-coupled double strand break repair (TC-DSBR) (*Chakraborty et al., 2015*; *Chakraborty et al., 2016*).

Here our data demonstrate that wtHTT is a part of a transcription-coupled DNA repair (TCR) complex formed by RNA polymerase II subunit A (POLR2A), basic transcription factors, PNKP, ATXN3, DNA ligase 3 (LIG 3), cyclic AMP response element-binding (CREB) protein (CBP, histone acetyltransferase), and this complex identifies lesions in the template DNA strand and mediates their repair during transcriptional elongation. The polyQ expansion in mHTT impairs PNKP and ATXN3 activities, disrupting the functional integrity of the TCR complex to adversely impact both transcription and DNA repair. Low PNKP activity leads to persistent accumulation of DNA lesions, predominantly in actively transcribing genes, resulting in unusual activation of the ATM-dependent p53 signaling pathway. Increased PNKP activity in mutant cells improved cell survival by substantially reducing DNA strand breaks and restricting ATM→p53 pathway activation. Likewise, low ATXN3 activity increases CBP ubiquitination and degradation thereby negatively influencing CREB-dependent transcription. These findings provide important mechanistic insights that could explain how mHTT may trigger neurotoxicity in HD.

## Results

### HTT is part of a TCR complex

Both wtHTT and mHTT interact with transcription factors and co-activators including CBP (*McCampbell et al., 2000*; *Nucifora et al., 2001*; *Steffan et al., 2000*), TATA-binding protein (TBP; *Huang et al., 1998*), p53 (*Bae et al., 2005*; *Steffan et al., 2000*), the general transcription factors TFIID and TFIIF (*Zhai et al., 2005*), and specificity protein 1 (Sp1; *Dunah et al., 2002*). POLR2A also interacts with HTT and is detected in nuclear inclusions in the HD brain (*Huang et al., 1998*; *Suhr et al., 2001*). It is hypothesized that wtHTT, which shuttles into the nucleus, assists in the assembly of transcription factor and co-activator complexes to regulate target gene expression, and that polyQ expansion perturbs the functional integrity of these complexes (*Kumar et al., 2014*; *Luthi-Carter and Cha, 2003*; *Ross and Tabrizi, 2011*). How mHTT disrupts the activities of specific promoters and whether mHTT-mediated transcriptional dysregulation is linked to DNA damage accumulation and aberrant DDR pathway activation remains unknown.

Given that HTT interacts with huntingtin-associated protein 1 (HAP-1; *Li et al., 1995*), while ATXN3 interacts with HAP-1 (*Takeshita et al., 2011*) and PNKP (*Chatterjee et al., 2015*; *Gao et al., 2015*), we asked whether ATXN3 and PNKP might interact with HTT to form a TCR complex and if this is affected by polyQ expansion. We isolated nuclear protein extract (NE) and cytosolic protein extract (CE) from SH-SY5Y cells and the fractions were analyzed by western blot (WB) to determine purity of nuclear protein fractions (*Figure 1A*). We immunoprecipitated (IP'd) endogenous wtHTT from the NE of SH-SY5Y cells, and WBs of the immunocomplexes (ICs) showed the presence of HAP-1, ATXN3, CBP, TAFII 130 (TAF4), POLR2A, PNKP, and LIG 3 (*Figure 1B* and *Figure 1—figure supplement 1*). Similarly, IP of endogenous ATXN3 from NEs revealed these proteins in the ATXN3-IC (*Figure 1C* and *Figure 1—figure supplement 2*). Finally, IP of PNKP from NEs confirmed that they were also present in the PNKP-IC (*Figure 1D* and *Figure 1—figure supplement 3*). To verify the specificity of these interactions in vivo, we analyzed the ICs for the presence of apurinic-apyrimidinic endonuclease 1 (APE1), another critical DNA base excision repair (BER) enzyme that works independently of PNKP-mediated BER pathways (*Wiederhold et al., 2004*). APE1 was not detected (*Figure 1B* to D), suggesting interaction specificity and selectivity. Finally, IP of POLR2A again revealed these proteins in the IC (*Figure 1E* and *Figure 1—figure supplement 4*). For further confirmation, we IP'd Myc-tagged HTT from the NEs of PC12 cells expressing Myc-tagged FL-wtHTT-Q23 or FL-mHTT-Q148. WB confirmed the presence of ATXN3, PNKP, POLR2A, CBP, and LIG three but not APE1 in the Myc-IC (*Figure 1F and G*, *Figure 1—figure supplements 5* and *6*), suggesting that HTT, POLR2A, CBP, ATXN3, LIG 3, and PNKP form a multiprotein TCR complex. Proximity ligation assays (PLAs) were then performed to verify interaction specificity (*Gao et al., 2015*). The reconstitution of fluorescence in neuronal cells (*Figure 1H* to M) and postmortem human brain sections (*Figure 1—figure supplement 7*) suggested substantial interaction among these proteins. Importantly, the majority of the PLA signals was from the nuclei but substantial amount of signals were from the periphery or cytoplasm. Immunostaining the cells with mitochondrial markers suggested that HTT

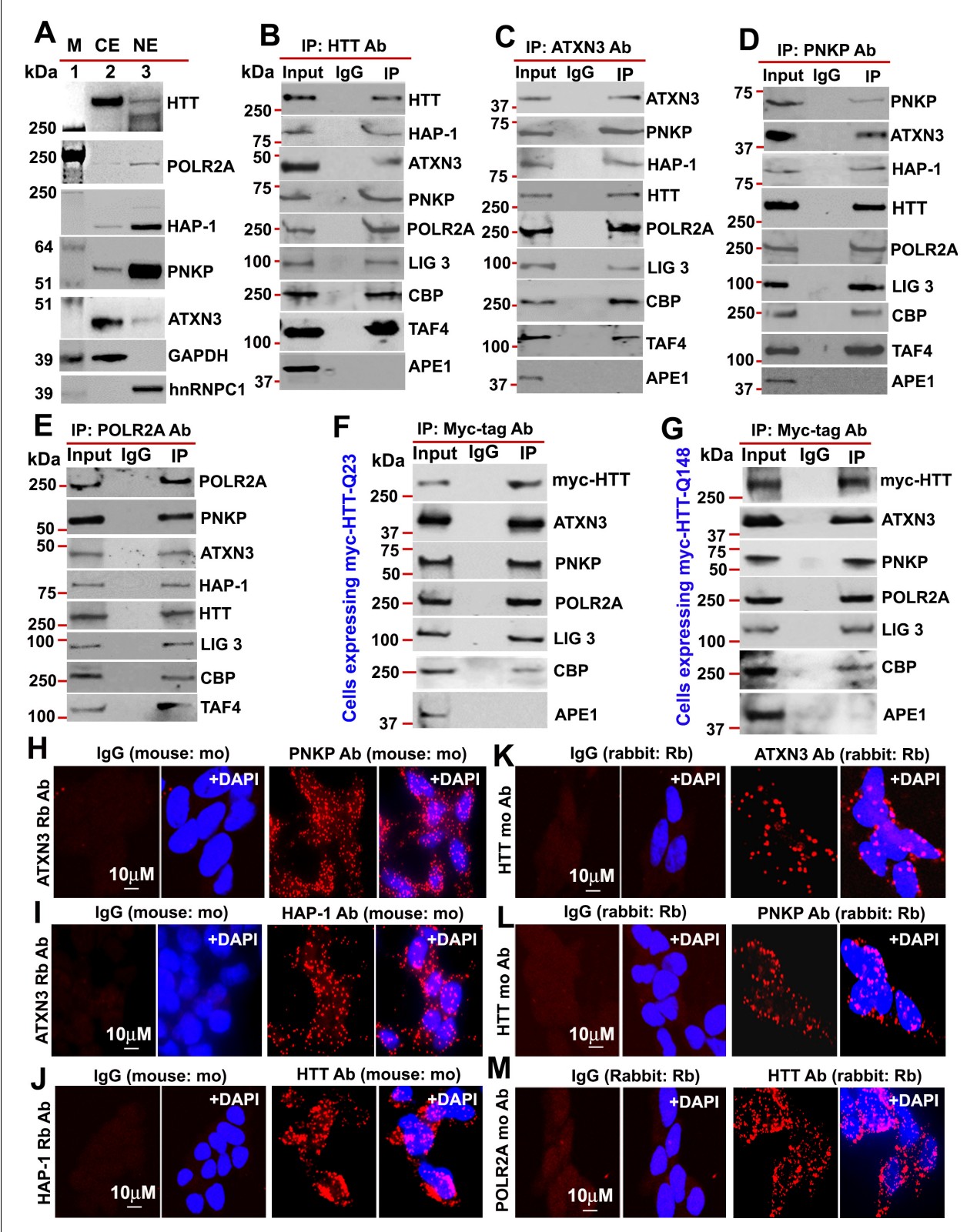

**Figure 1.** HTT is a part of the TCR complex. (**A**) Nuclear extract (NE), and cytosolic extract (CE) were purified from human neuroblastoma SH-SY5Y cells and the protein fractions were analyzed by western blots (WBs) to detect HTT, ATXN3, PNKP, and HAP1 levels in these sub-cellular fractions. GAPDH and hnRNPC1/C2 were used as cytosolic and nuclear markers respectively. APE1 was used as a negative control in panels A to D. (**B**) Endogenous HTT was immunoprecipitated (IP'd) from NEs of SH-SY5Y cells and immunocomplex (IC) were analyzed by western blot (WB) to examine the TCR proteins

*Figure 1 continued on next page*

Figure 1 continued

(HAP-1, ATXN3, PNKP, POLR2A, LIG 3, CBP, and TAFII 130 (TAF4). (**C**) Endogenous ATXN3 was IP'd from NEs of SH-SY5Y cells and IC was subjected to WB to detect associated TCR complex components with respective antibodies. (**D**) Endogenous PNKP was IP'd from NEs of SH-SY5Y cells and IC was analyzed by WB to examine associated TCR components. (**E**) Endogenous POLR2A was IP'd from NEs of SH-SY5Y cells and IC was analyzed by WB to detect associated TCR proteins. (**F**) NEs was isolated from PC12 cells ectopically expressing a Myc-tagged full-length normal wild type HTT (FL-wtHTT-Q23) for assessing the possible interaction of HTT with POLR2A and associated TCR proteins. Exogenous Myc-wtHTT-Q23 was IP'd with an anti-Myc antibody, and the Myc immunocomplex was subjected to WBs with respective antibodies. APE1 was used as a negative control in panels F and G. (**G**) NEs was isolated from PC12 cells ectopically expressing a Myc-tagged full-length mutant HTT (FL-mHTT-Q148). Exogenous Myc-wtHTT-Q148 was IP'd with an anti-Myc antibody, and the Myc immunocomplex was subjected to WBs with respective antibodies. Proximity Ligation Assay (PLA) in SH-SY5Y cells to examine the protein-protein interaction using the following antibody pairs. Red fluorescence indicates positive PLA signals for protein-protein interactions. Nuclei were stained with DAPI. (**H**) ATXN3 (rabbit: Rb) and IgG (mouse: mo) or PNKP (mouse: mo) antibodies. (**I**) ATXN3 (rabbit: Rb) with IgG (mouse: mo) or HAP-1 (mouse: mo) antibodies. (**J**) HAP-1 (rabbit: Rb) and IgG (mouse: mo) or HTT (mouse: mo) antibodies. (**K**) HTT (mouse: mo) with IgG (rabbit: Rb) or ATXN3 (rabbit: Rb) antibodies. (**L**) HTT (mouse: mo) with IgG (rabbit: Rb) or PNKP (rabbit: Rb) antibodies, and. (**M**) POLR2A (mouse: mo) with IgG (rabbit: Rb) or HTT (rabbit: Rb) antibodies.

DOI: https://doi.org/10.7554/eLife.42988.003

The following figure supplements are available for figure 1:

**Figure supplement 1.** Huntingtin is a part of the transcription-coupled DNA repair (TCR) complex.
DOI: https://doi.org/10.7554/eLife.42988.004

**Figure supplement 2.** ATXN3 is present in the HTT-TCR complex.
DOI: https://doi.org/10.7554/eLife.42988.005

**Figure supplement 3.** DNA strand break repair enzyme PNKP is present in the TCR complex.
DOI: https://doi.org/10.7554/eLife.42988.006

**Figure supplement 4.** RNA polymerase large subunit (POLR2A) is present in the HTT-TCR complex.
DOI: https://doi.org/10.7554/eLife.42988.007

**Figure supplement 5.** Wild-type normal HTT is a part of the multiprotein TCR complex in vitro.
DOI: https://doi.org/10.7554/eLife.42988.008

**Figure supplement 6.** Mutant HTT is a part of the multiprotein TCR complex in vitro.
DOI: https://doi.org/10.7554/eLife.42988.009

**Figure supplement 7.** HTT is a component of the TCR complex in vivo.
DOI: https://doi.org/10.7554/eLife.42988.010

forms similar complexes in the mitochondria (data not shown). Importantly, about 60–70% of the PLA signal was nuclear in control brain, while the complexes were predominantly in the perinuclei or cytoplasm of HD brain sections (*Figure 1—figure supplement 7*). Since PNKP and HTT are present in the mitochondria (*Mandal et al., 2012*; *Orr et al., 2008*), the extranuclear signals detected in the control subjects are presumably from mitochondrial HTT-ATXN3-PNKP complexes. WB analysis of subcellular protein fractions from neuronal cells show the presence of HTT, ATXN3 and PNKP in mitochondria (data not shown). Moreover, co-staining the cells or brain sections with mitochondrial markers suggested presence of HTT, ATXN3 and PNKP in mitochondria (data not shown). These findings indicate that HTT may form a similar complex in mitochondria regulating mtDNA repair and transcription.

The possible in vivo association of these proteins was further assessed by immunostaining HTT, PNKP, and ATXN3 in postmortem brain tissue from patients with HD and control subjects. Confocal microscopy revealed colocalization of HTT with PNKP and ATXN3 in HD and control brain (*Figure 2A & B*; arrows). Colocalization of ATXN3 with PNKP was observed in both groups (*Figure 2C*; arrows). Marked HTT/PNKP colocalization was also observed in brain sections from HD knock-in (zQ175; *Menalled et al., 2012*) and WT control mouse brain tissue (data not shown).

## The C-terminal catalytic domain of PNKP interacts with HTT

PNKP contains an N-terminal fork head-associated (FHA) domain, C-terminal fused 3'-phosphatase (PHOS) domain, and 5'-kinase (KIN) domain. The PHOS domain hydrolyzes 3'-phosphate groups, while the KIN domain promotes addition of a phosphate group to the 5'-OH at damaged sites for error-free repair (*Karimi-Busheri et al., 1999*). To identify the specific PNKP domain(s) that interact with HTT, full-length PNKP (FL-PNKP); the FHA, PHOS, and KIN domains; the FHA + PHOS domains; or the PHOS + KIN domains were expressed as a FLAG-tagged peptide, as illustrated in *Figure 3A*. We individually expressed these domains in SH-SY5Y cells (*Figure 3B*; upper panel) and isolated the

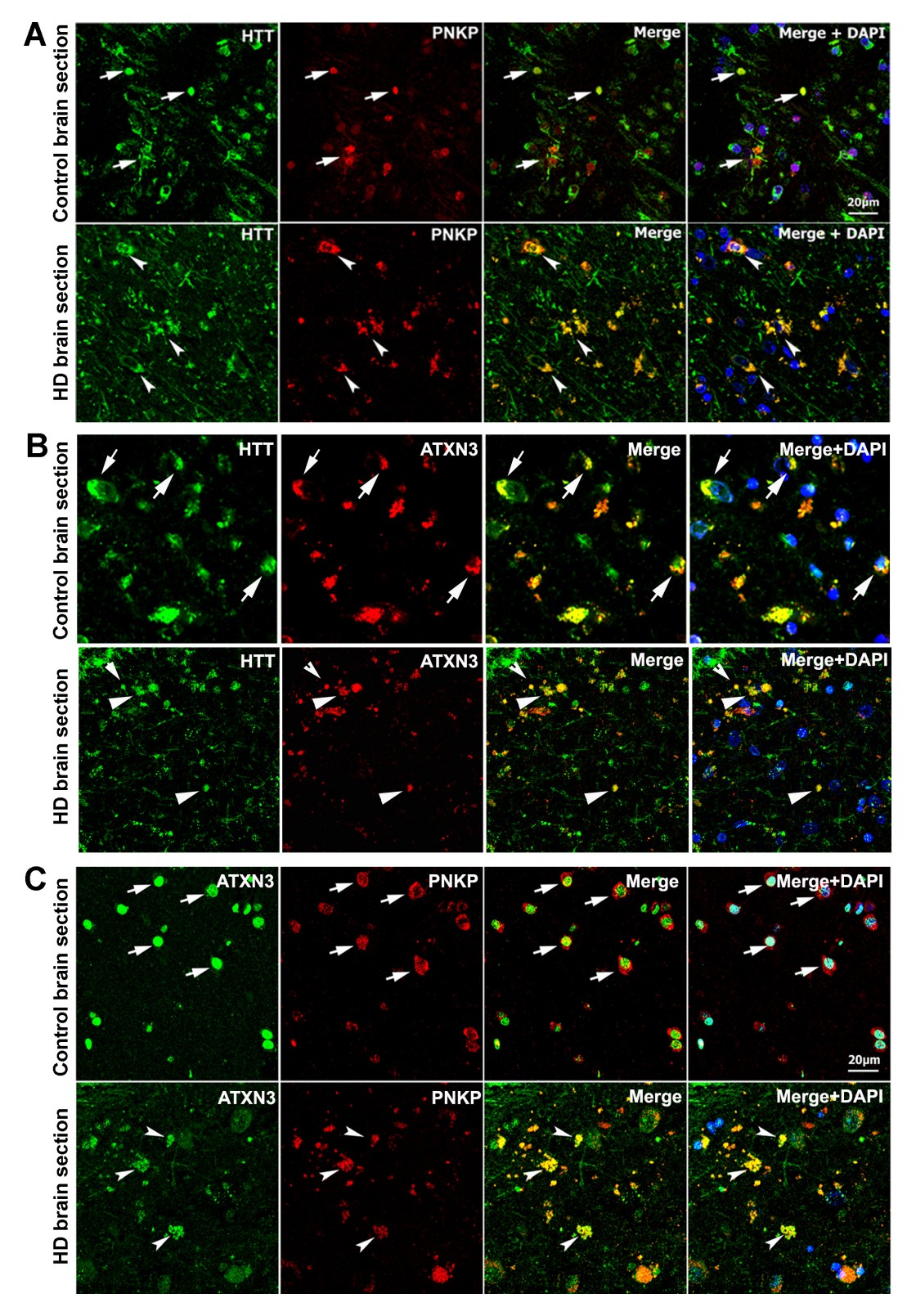

**Figure 2.** HTT colocalizes with PNKP and ATXN3 in postmortem human brain sections. (**A**) Normal and HD postmortem brain (mHTT-Q94) sections were analyzed by double immunolabeling with antibodies against HTT (green) and PNKP (red) to assess their in vivo colocalization and possible interactions (representative colocalization of HTT and PNKP are shown by arrows). For panels A and B, merge of red and green fluorescence appears as yellow/orange, and nuclei were stained with DAPI (blue). (**B**) Normal and HD brain (mHTT-Q82; early onset HD patients, disease grade 4/4, manifesting

*Figure 2 continued on next page*

Figure 2 continued

severe phenotype) sections analyzed by double immunolabeling with antibodies against HTT (green) and ATXN3 (red) to assess their in vivo colocalization and possible interaction (arrows). (C) Normal and HD brain (mHTT-Q94; early onset HD patients, disease grade 4/4, manifesting severe phenotype) sections were analyzed by double immunolabeling with antibodies against ATXN3 (green) and PNKP (red) to assess their in vivo colocalization and possible interaction (arrows).

DOI: https://doi.org/10.7554/eLife.42988.011

NEs. IPs of these domains with a FLAG antibody and subsequent WB analysis of the IC showed the presence of HTT in the FLAG-(FL-PNKP)-IC and FLAG-(PHOS + KIN)-IC (*Figure 3B*; Lower panel, lanes 1 and 6, arrow). HTT was not detected in FLAG-ICs when the individual FHA, PHOS, or KIN domains were IP'd (*Figure 3B*; lanes 2–5, arrow). This suggests that the C-terminal catalytic domain of PNKP interacts with HTT, but the individual FHA, PHOS, and KIN domains are not sufficient. We separately expressed the PNKP domains in cells, isolated the NEs, and IP'd endogenous HTT. WBs revealed the presence of full-length and PHOS + KIN domains in the HTT-IC (*Figure 3C*; Lower panel, lanes 1 and 6). When we expressed the PNKP domains in PC12 cells expressing Myc-wtHTT-Q23 or Myc-mHTT-Q148 (*Figure 3D & E*; upper panels), IP of the Myc-HTT and WB revealed the full-length protein or PHOS + KIN domain (*Figure 3D & E*, Lower panels, lanes 1 and 6). These data suggest that both wtHTT and mHTT interact with the C-terminal catalytic domain of PNKP.

## N-terminal-truncated HTT fragments interact with the catalytic domain of PNKP

The N-terminal-truncated fragment of mHTT (NT-mHTT) containing the polyQ expansion is encoded by exon 1 of the HTT gene. Transgenic mice expressing exon one or a truncated fragment extending beyond the first exon (N171) with NT-mHTT recapitulate HD-like neurological and behavioral abnormalities (*Mangiarini et al., 1996*; *Schilling et al., 1999*). To test whether this fragment interacts with PNKP, we expressed NT-wtHTT-Q23 or NT-mHTT-Q97 (1–586 base pairs) as a GFP-tagged peptide in SH-SY5Y cells (*Figure 4A*; upper panel), isolated the NEs, and IP'd the GFP-NT-HTT fusion peptide with a GFP antibody. WBs showed the presence of PNKP, ATXN3, and HTT in the GFP-IC (*Figure 4A*; lower panel, lanes 4 and 6). We next IP'd this fragment from PC12 cells expressing Myc-NT-wtHTT-Q23 or NT-mHTT-Q148 and found ATXN3, PNKP, POLR2A, CBP, and LIG three in the Myc-IC. Importantly, APE1 was not detected in the IC, again suggesting interaction specificity (*Figure 4B*). To identify which PNKP domain interacts with NT-HTT, we expressed various domains as FLAG-tagged peptides in SH-SY5Y cells expressing either Myc-NT-HTT-Q23 or Myc-NT-HTT-Q97 (*Figure 4C & D*; upper panels) and IP'd Myc-tagged fragments from the NEs. WBs revealed FL-PNKP and PNKP-(PHOS + KIN) domains in the Myc immunocomplex (*Figure 4C & D*; lanes 1 and 6), suggesting that the N-terminal fragment of HTT interacts with the C-terminal catalytic domain of PNKP. However, from the WB analyses we could not establish whether the interaction of the mutant HTT fragment (NT-mHTT) with PNKP-(PHOS + KIN) domain is stronger than the interaction with the N-terminal fragment of WT HTT (NT-wtHTT; *Figure 4C & D*; lanes 6). The PNKP-(FHA + PHOS) domain also showed a relatively weaker interaction with the N-terminal fragment of HTT (*Figure 4C & D*; lanes 5) indicating that the FHA-PHOS domain of PNKP alone interacts with the N-terminal fragment of HTT.

To further assess these possible interactions, we performed bi-molecular fluorescence complementation (BiFC) assays as we previously reported (*Gao et al., 2015*). We cloned either the full-length or C-terminal catalytic domain of PNKP at the N-terminus of cyan fluorescent protein (CFP) into plasmid pBiFC-VN173 to construct plasmids pVN-PNKP and pVN-(PHOS +KIN), respectively. We also cloned the N-terminal fragment of wtHTT and mHTT cDNA (encoding 23 and 97 glutamines, respectively) at the C-terminus of CFP in plasmid pBiFC-VC155 to construct pVC-NT-HTT-Q23 and pVC-NT-HTT-Q97, respectively (detailed descriptions of these plasmids are provided in the STAR Methods). Cotransfection of plasmid pVN-PNKP with the parent plasmid pBIFC-VC155 did not reconstitute fluorescence (*Figure 4E*; Panel 1), whereas cotransfection of pVN-PNKP with either pVC-NT-HTT-Q23 or pVC-NT-HTT-Q97 did (*Figure 4E*; Panels 2 and 3). Similarly, cotransfection of pVN-(PHOS +KIN) with pBIFC-VC155 did not produce fluorescence (*Figure 4E*; Panel 4), whereas cotransfection of pVN-(PHOS +KIN) with either pVC-NT-HTT-Q23 or pVC-NT-HTT-Q97 robustly

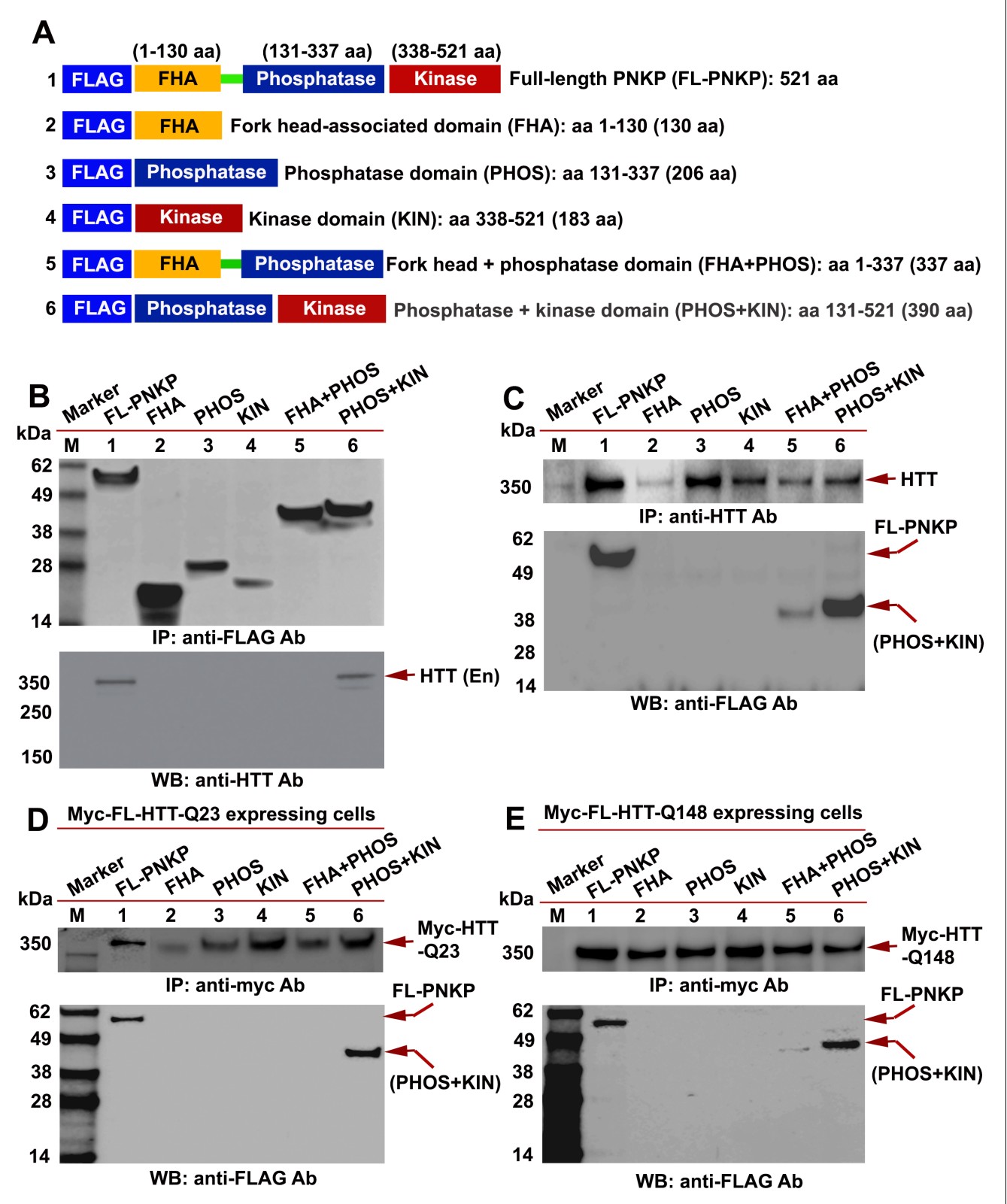

**Figure 3.** HTT interacts with the C-terminal catalytic domain of PNKP. (**A**) Schematic illustrating various functional domains of PNKP expressed as FLAG-tagged peptides: (1) full-length PNKP containing N-terminal fork-head-associated (FHA) domain, central phosphatase (PHOS) and C-terminal kinase (KIN) domains; (2) FHA domain (1–130 amino acids); (3) PHOS domain (131–337 amino acids); (4) KIN domain (338–521 amino acids); (5) FHA and PHOS domains (1–137 amino acids); and (6) PHOS and KIN domains (131–521 amino acids). (**B**) Plasmids encoding full-length PNKP (FL-PNKP) or

*Figure 3 continued on next page*

*Figure 3 continued*

various PNKP domains were separately transfected into SH-SY5Y cells (lanes 1 to 6) and NEs isolated 48 hr post-transfection. Lanes 1 to 6 in the WB (upper panel) shows the pull-down of full-length PNKP (FL-PNKP) and various PNKP domains that were IP'd with an anti-FLAG Ab. The WB in the lower panel shows the presence of endogenous HTT (arrow) in the FLAG-IC. M: protein molecular weight marker. (C) Plasmids encoding full-length PNKP (FL-PNKP) and various PNKP domains were separately transfected into SH-SY5Y cells (lanes 1 to 6), NEs isolated, and HTT was IP'd with an anti-HTT antibody. The pull-down of endogenous HTT is shown in the upper panel (arrow). The HTT-IC was analyzed by WB (lower panel) to detect FL-PNKP or various PNKP domains with an anti-FLAG Ab (arrows). (D) Plasmids encoding FLAG-tagged full-length PNKP (FL-PNKP) or various PNKP domains were transfected separately into PC12 cells expressing full-length Myc-tagged normal HTT encoding 23Qs (Myc-FL-wtHTT-Q23) (lanes 1 to 6), NEs were isolated, and Myc-HTT IP'd with Myc tag antibody. Upper panel is the WB showing the IP of HTT with an anti-Myc tag antibody. The Myc-IC was analyzed by WB to assess interaction of various PNKP domains with HTT with an anti-FLAG antibody (lower panel, arrows). (E) Plasmids encoding the full-length PNKP (FL-PNKP) or various PNKP domains were transfected into PC12 cells expressing Myc-tagged full-length mutant HTT encoding 148Qs (Myc-FL-mHTT-Q148) (lanes 1 to 6), NEs isolated, and Myc-tagged HTT was IP'd with an anti-Myc-tag antibody. Upper panel is the WB showing IP of Myc-HTT with anti-Myc tag antibody. Interactions of FL-PNKP or various PNKP domains with FL-HTT were analyzed by WB with an anti-FLAG antibody (lower panel, arrows).

DOI: https://doi.org/10.7554/eLife.42988.012

reconstituted fluorescence (*Figure 4E*, Panels 5 and 6). Although these data suggest that the N-terminal of mHTT interacts with PNKP, these experiments do not inform the relative strengths of interaction between these peptides. Nonetheless, the IP and BIFC studies suggest that the truncated-N-terminal fragments of both WT and mHTT interact with the C-terminal catalytic domain of PNKP. The interaction of these peptides with the PHOS-KIN domain of PNKP is relatively stronger than with the PHOS domain alone. However, more rigorous structural and biophysical measurements using purified proteins/peptides will be required to understand the true nature of these protein-protein interactions, the relative binding efficacies and to identify the direct interacting partners in this complex. Moreover, since the HTT-TCR complex is not fully characterized, the presence of additional unidentified components of the complex could significantly alter these interactions in vivo.

## mHTT abrogates PNKP activity to induce DNA damage and trigger DDR signaling

Given that PNKP interacts with mHTT, we measured the 3'-phosphatase activity of PNKP in induced pluripotent stem cells (iPSCs) differentiated to neurons enriched for medium striatal neuronal populations from HD and unaffected control subjects using a modification of *Telezhkin et al. (2016)*. HD iPSC-derived neurons (mHTT-109Qs) were compared to control neurons (wtHTT-33Q; HD iPSC *HD iPSC Consortium, 2017*) and activity was found to be 70–80% lower in the NE of HD neurons, while PNKP protein levels did not change (representative experiment, *Figure 5A* to C). Similar differences were found for neurons with adult onset alleles (Q50 and Q53) compared to controls (Q18 and Q28). In these comparisons there was substantially reduced (70% to 80%) PNKP activity in HD neurons (Q50 and Q53) compared with control neurons (Q18 or Q28) (*Figure 5—figure supplement 1*), supporting an impairment in human neurons in the presence of mHTT.

We next measured PNKP activity in PC12 cells expressing exogenous full-length wtHTT (FL-wtHTT-Q23) and full-length mHTT (FL-mHTT-Q148) (*Igarashi et al., 2003*; *Tanaka et al., 2006*). We found that it was about 30–40% higher in the NE of PC12 cells expressing wtHTT, and about 70% lower in the NE of PC12 cells expressing FL-mHTT-Q148 compared to control cells, while PNKP protein levels did not change (*Figure 5—figure supplement 2A* to C). These data suggest that wtHTT and mHTT stimulate and abrogate PNKP activity, respectively. Since wtHTT interacts with and stimulates PNKP activity, we examined the extent to which HTT depletion alters PNKP activity. We found that in HTT-depleted cells, PNKP activity was reduced by >70% (*Figure 5—figure supplement 2D* to G), suggesting that wtHTT plays key roles in stimulating PNKP activity, maintaining the functional integrity of the TCR complex, and repairing DNA damage. PNKP activity was 80–90% decreased in the striatum (STR) and cortex (CTX), and marginally (5%) decreased in the cerebellum (CRBL) of male heterozygous asymptomatic zQ175 mice at 7 weeks, whereas PNKP protein levels were not different from WT (*Figure 5D* to F). An identical trend was observed in female littermates (data not shown).

Because the N-terminal of mHTT interacts with PNKP, we investigated whether N-terminal truncated fragment of mHTT interferes with PNKP activity in PC12 cells or N171-82Q mice (*Schilling et al., 1999*; *Tanaka et al., 2006*). PNKP activity was ~30% higher in PC12 cells expressing

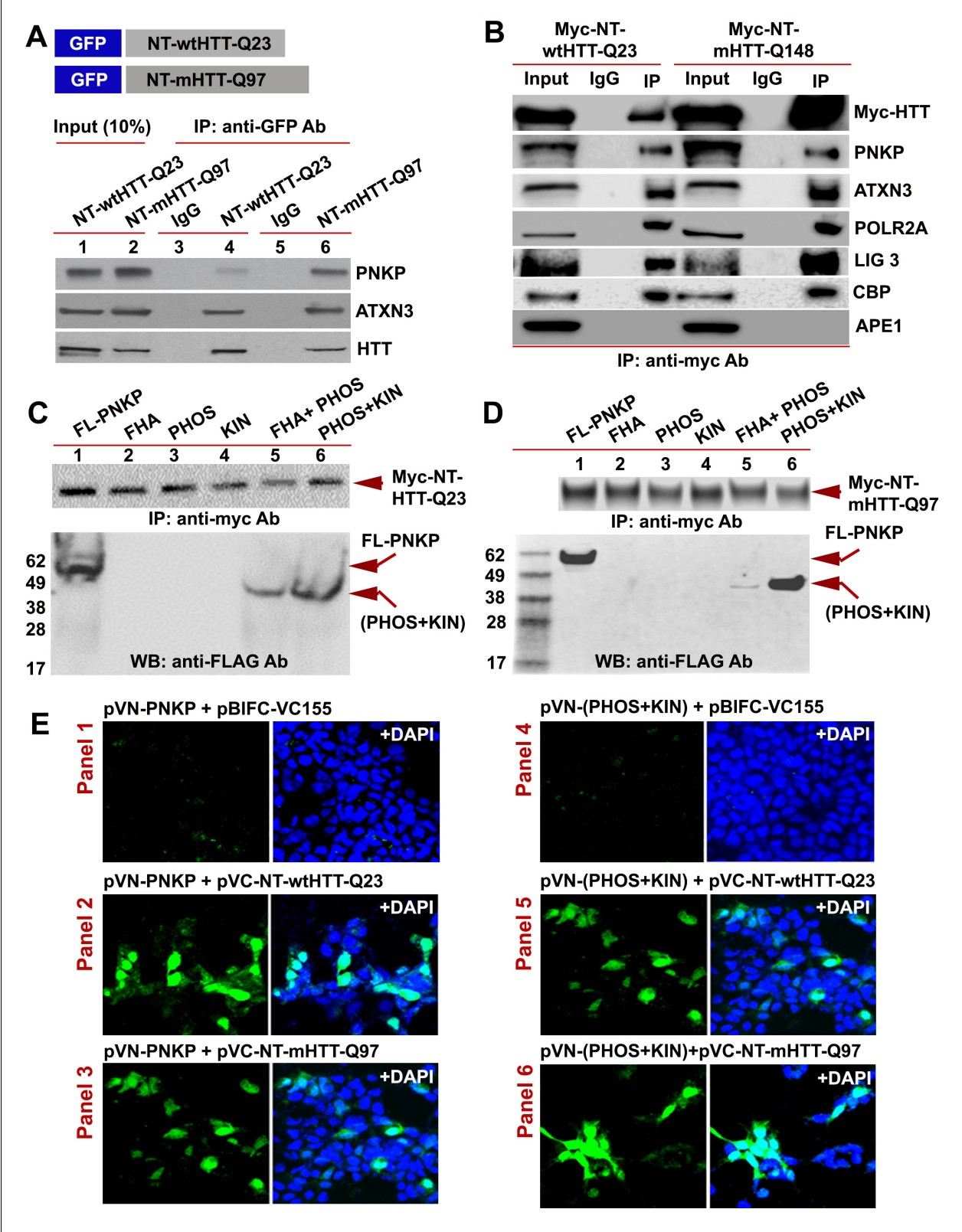

**Figure 4.** The N-terminus of HTT interacts with the C-terminal catalytic domain of PNKP. (A) Schematic showing GFP-tagged N-terminal fragment of wild-type normal HTT encoding 23Qs or mutant HTT encoding 97Qs (NT-wtHTT-Q23 and NT-mHTT-Q97 plasmid vectors respectively; upper panel). SH-SY5Y cells were transfected with NT-wtHTT-Q23 or NT-mHTT-Q97, NEs isolated, fusion peptides IP'd from NE with an anti-GFP antibody, and WBs performed with respective antibodies to detect endogenous PNKP, ATXN3, or HTT in the GFP-IC (lower panel). (B) NEs from PC12 cells expressing

*Figure 4 continued on next page*

*Figure 4 continued*

Myc-tagged N-terminal fragment of wild-type normal HTT encoding 23Qs or mutant HTT encoding 148Qs (NT-wtHTT-Q23 or NT-mHTT-Q148, respectively) were isolated and the Myc-HTT was IP'd with an anti-Myc tag Ab and Myc-IC was analyzed by WBs to detect various TCR complex components with respective antibodies. (**C**) Plasmids encoding full-length PNKP (FL-PNKP) or various PNKP domains (lanes 1 to 6) were separately transfected into SH-SY5Y cells expressing the N-terminal fragment of HTT encoding 23Qs (Myc-NT-wtHTT-Q23), NEs were isolated and the NT-HTT was IP'd with an anti-Myc tag Ab. The upper panel shows pull down of Myc-NT-HTT-Q23. The Myc-IC was analyzed by WBs with an anti-FLAG Ab to detect FL-PNKP or PNKP domains (lower panel; arrows). (**D**) Plasmids encoding full-length PNKP (FL-PNKP) or various domains (lanes 1 to 6) were separately transfected into SH-SY5Y cells expressing the N-terminal fragment of mutant HTT encoding 97Qs (Myc-mHTT-Q97) and NEs were isolated and the Myc-NT-HTT-Q97 was IP'd with an anti-Myc tag Ab and the Myc-IC was analyzed by WBs to detect FL-PNKP or PNKP domains (lower panel; arrows). (**E**) BiFC assay of SH-SY5Y cells cotransfected with plasmids: Panel 1) pVN173-PNKP and pVC-BIFC-155, Panel 2) pVN-PNKP and pVC-NT-wtHTT-Q23, Panel 3) pVN-PNKP and pVC-NT-mHTT-Q97, Panel 4) pVN (PHOS + KIN) and pVC-BIFC-155, Panel 5) pVN-(PHOS + KIN) and pVC-NT-wtHTT-Q23, and panel 6) pVN-(PHOS + KIN) and pVC-NT-mHTT-Q97. Reconstitution of fluorescence was monitored via fluorescence microscopy. Nuclei were stained with DAPI (blue).
DOI: https://doi.org/10.7554/eLife.42988.013

NT-wtHTT-Q23 and >80% lower in cells expressing NT-mHTT-Q148 (*Figure 5—figure supplement 3A & B*). Similar to full-length HTT, PNKP activity was decreased in SH-SY5Y cells expressing NT-mHTT with variable glutamine expansions (*Figure 5—figure supplement 3C & D*). Moreover, PNKP activity was >80% decreased in the STR and CTX of N171-82Q brain compared to control (*Figure 5—figure supplement 3E & F*). To test if mHTT specifically blocks PNKP activity rather than interfering with DNA repair per se, we examined how it modulated the repair of two nicked DNA duplexes: one without a 3'-phosphate end that requires DNA polymerase and ligase activities but not PNKP activity for repair, and another duplex with a 3'-phosphate end that requires PNKP and DNA polymerase and ligase activities for complete repair. We observed that NEs from cells expressing mHTT or from zQ175 mouse brain did not hamper repair of the duplex that required DNA polymerase and ligase activities but did not require PNKP activity. In contrast, NEs from these cells and mice did abrogate repair of the duplex that requires PNKP (*Figure 5G* to J), suggesting that mHTT specifically blocks PNKP activity but does not interfere with the activities of other repair enzymes in the TCR complex. In response to DNA strand break accumulations, ATM is activated by phosphorylation which phosphorylates p53, which in turn activates pro-apoptotic gene transcription (*Chipuk et al., 2004*; *Nakano and Vousden, 2001*; *Oda et al., 2000*). Consistently, we found chronic activation of the DDR-ATM-p53 pathway in HD neurons *Figure 5—figure supplement 4A & B*) and in zQ175 CTX (*Figure 5—figure supplement 4C & D*) compared with respective controls. mHTT expression has been shown to activate p53 in HD, whereas deleting p53 in the HD transgenic brain rescues behavioral abnormalities (*Bae et al., 2005*). Consistently, markedly increased mRNA expression of p53 target genes (e.g., Bcl2L11, Pmaip1, Bid, Pidd1 and Apaf1) were observed in the STR but not in CRBL of zQ175 mice compared to controls (*Figure 5—figure supplement 4E & F*).

We next expressed the N-terminal truncated fragment of mHTT encoding Q97 (NT-mHTT-Q97) in SH-SY5Y cells overexpressing PNKP and carried out a comet assay (*Olive and Banáth, 2006*). Analysis of mutant cells showed more strand breaks, which were substantially rescued after PNKP overexpression (*Figure 5—figure supplement 5A* to C), suggesting that mHTT-mediated ablation of PNKP activity contributes to increased DNA strand breaks. Consistently, we noted activation of ATM-p53 signaling in cells expressing NT-mHTT-Q97 (*Figure 5—figure supplement 5D & E*), and PNKP overexpression reduced mHTT-mediated DDR-ATM pathway activation (*Figure 5—figure supplement 5F*). PC12 cells expressing the full-length mHTT encoding 148Qs (FL-mHTT-Q148) showed increased caspase-3 activity and PNKP overexpression reduced caspase-3 activation *Figure 5—figure supplement 5G*). Consistently, PC12 cells expressing FL-mHTT-Q148 also showed higher cell toxicity and PNKP overexpression significantly rescued cell toxicity (*Figure 5—figure supplement 5H & I*). Collectively, these results suggest that mHTT-mediated activation of the ATM-p53 pathway and associated cell toxicity is at least partially due to PNKP inactivation by mHTT.

## mHTT preferentially induces DNA breaks in the transcriptionally active genome

Emerging evidence suggests that the TCR complex plays a pivotal role in editing strand breaks in actively transcribing template DNA to maintain genome integrity and cell survival, and its

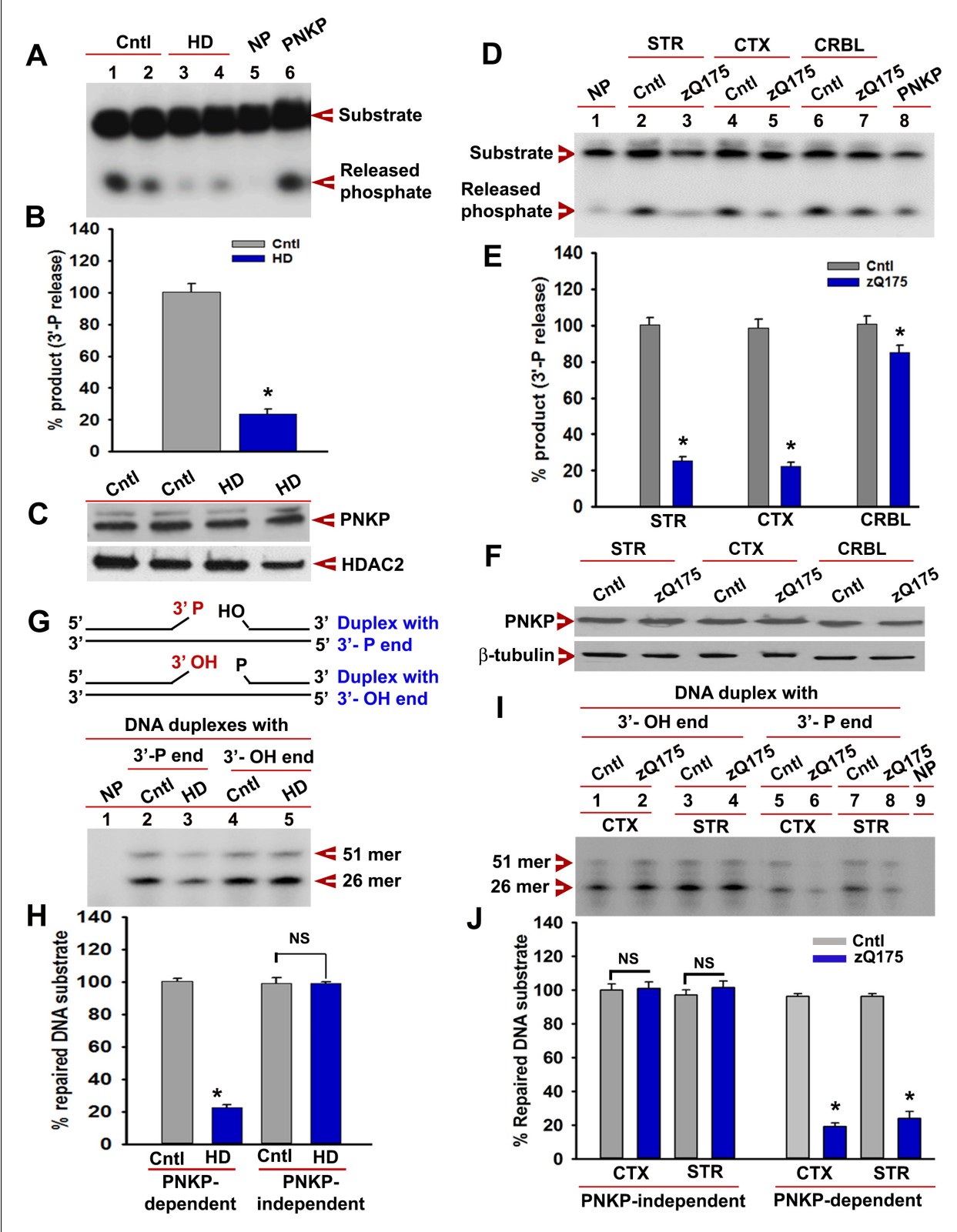

**Figure 5.** mHTT abrogates PNKP activity in vitro and in vivo. (**A**) The 3'-phosphatase activities of PNKP in the NE (250 ng each) of control (lanes 1 and 2, differentiation replicates of Q33 iPSCs), and HD neurons (lanes 3 and 4, differentiation replicates of Q109 iPSCs) were determined by amount of phosphate release from the DNA substrate (arrows). No protein extract was added to the substrate in lane 5 (NP), and purified PNKP (25 fmol) was added as a positive control (lane 6). (**B**) Relative 3'-phosphatase activities (in terms of % product) of PNKP in control (Q33) and HD (Q109) neurons. Data

*Figure 5 continued on next page*

*Figure 5 continued*

represent mean ± SD, *p<0.001 when compared with control. The quantification was measured by taking into account two biological replicates and three technical replicates. (C) NEs from control (Q33) and HD neurons (Q109) were analyzed by WB to determine PNKP protein levels (upper panel); HDAC2 was used as a loading control (lower panel). (D) PNKP activities in the NEs from the striatum (STR), cortex (CTX) and cerebellum (CRBL) of 7 weeks old WT control and zQ175 transgenic mice (n = 5; STR or CTX or CRBL were pooled from five littermate mice); no protein was added to the substrate in lane 1 (NP), and purified PNKP was added as a positive control (lane 8). (E) Relative PNKP activities (in terms of % product) in the STR, CTX, and CRBL of 7 weeks old zQ175 transgenic (n = 5) and age-matched control (n = 5) mice. Five biological replicates and three technical replicates were used in this study. Data represent mean ± SD, *p<0.001 when compared with control. (F) NEs from the STR, CTX, and CRBL of zQ175 (n = 5) and age-matched wild type control (n = 5) mice were analyzed by WB to determine PNKP levels (upper panel); β-tubulin was used as a loading control (lower panel). (G) mHTT specifically abrogates PNKP activity without interfering with DNA polymerase or ligase activities. Total DNA repair was assessed with NE (2.5 µg) from control (Q33) and HD (Q109) neurons added to two nicked DNA duplexes (upper panel): one with 3′-phosphate ends that require PNKP activity (lanes 2 and 3, lower panel), and the other with clean 3′-OH termini that do not require PNKP activity but need DNA polymerase and ligase activities (lanes 4 and 5, lower panel) for effective repair. The 51-mer DNA band (arrow) represents repaired DNA duplexes in G and I. (H) Relative PNKP and PNKP-independent DNA repair efficiencies in HD (Q109) and control (Q33) primary neurons. NS denotes not significant difference in H and J. Two biological replicates and three technical replicates were used in this study. Data represent mean ± SD. (I) NEs from zQ175 transgenic (n = 5) and control (n = 5) mice CTX and STR were added to nicked DNA substrates as described above, and total DNA repair was assessed. (J) PNKP-dependent or -independent repair of the DNA duplexes by NEs from control and zQ175 transgenic mouse brain tissue. Data represent mean ± SD, *p<0.001 for E, H, and J. Three biological replicates and three technical replicates were used in this assay.

DOI: https://doi.org/10.7554/eLife.42988.014

The following source data and figure supplements are available for figure 5:

**Source data 1.** Huntinton's disease models.
DOI: https://doi.org/10.7554/eLife.42988.020

**Figure supplement 1.** Endogenous level of mHTT is sufficient to deplete nuclear PNKP activity in iPSC-derived HD primary striatal neurons.
DOI: https://doi.org/10.7554/eLife.42988.015

**Figure supplement 2.** mHTT-mediated inactivation of the TCR complex abrogates PNKP activity.
DOI: https://doi.org/10.7554/eLife.42988.016

**Figure supplement 3.** Expression of the N-terminus of mHTT abrogates PNKP activity.
DOI: https://doi.org/10.7554/eLife.42988.017

**Figure supplement 4.** mHTT triggers DNA damage response (DDR)-ATM signaling.
DOI: https://doi.org/10.7554/eLife.42988.018

**Figure supplement 5.** PNKP overexpression in mutant cells rescues cell toxicity.
DOI: https://doi.org/10.7554/eLife.42988.019

inactivation leads to preferential accumulation of DNA breaks in the transcriptionally active genome (*Chakraborty et al., 2015*; *Chakraborty et al., 2016*; *Hanawalt and Spivak, 2008*). Since mHTT abrogates the activity of PNKP, a key component of the TCR complex (*Chakraborty et al., 2016*), we compared the associations of HTT and TCR proteins with transcriptionally active versus inactive genomes and asked whether the former accumulates more strand breaks in the HD brain. Chromatin immunoprecipitation (ChIP) revealed significantly higher HTT occupancy on actively transcribing genes in the brain (e.g., neuronal differentiation factor 1 and 2 [Neurod1 and Neurod2], neurogenic basic-helix-loop-helix protein neurogenin 1 [Neurog1], tubulin beta three class III [Tubb3], neuron-specific enolase 2 [Eno2γ], and DNA polymerase beta [Pol b]) over genes that are not transcribed in the brain but actively transcribed in skeletal or cardiac muscle (e.g., myogenic differentiation factor 1 [Myod1]; myogenic factor 4; myogenin [Myog]; and myosin heavy chain 2, 4, 6, or 7 [Myh2, Myh4, Myh6, or Myh7]; (*Figure 6A & B*). Increased association between HTT with the transcriptionally active genome and mHTT-mediated abrogation of PNKP activity indicate that the wtHTT-TCR complex repairs lesions during transcriptional elongation, but polyQ expansion might impair the TCR and facilitate DNA damage accumulation. To test this theory, we performed Long-amplicon quantitative polymerase chain reaction (LA-qPCR) analysis, a versatile technique to measure nuclear and mitochondrial DNA damage (*Gao et al., 2015*; *Santos et al., 2006*) to assess DNA strand breakage in actively transcribing and non-transcribing genes in the transgenic mouse cortex (CTX). The results revealed 60–70% lower PCR-amplification of actively transcribing genes in asymptomatic (seven wks) zQ175 mouse CTX compared to age-matched WT controls (*Figure 6C & D*). In contrast, the amplification efficacy for non-transcribing genes in the zQ175 CTX was only marginally (10–15%) reduced (*Figure 6E & F*), indicating less DNA damage accumulation. Consistent with the levels of PNKP activities observed in the striatum (STR) and cerebellum (CRBL), the LA-qPCR analysis revealed

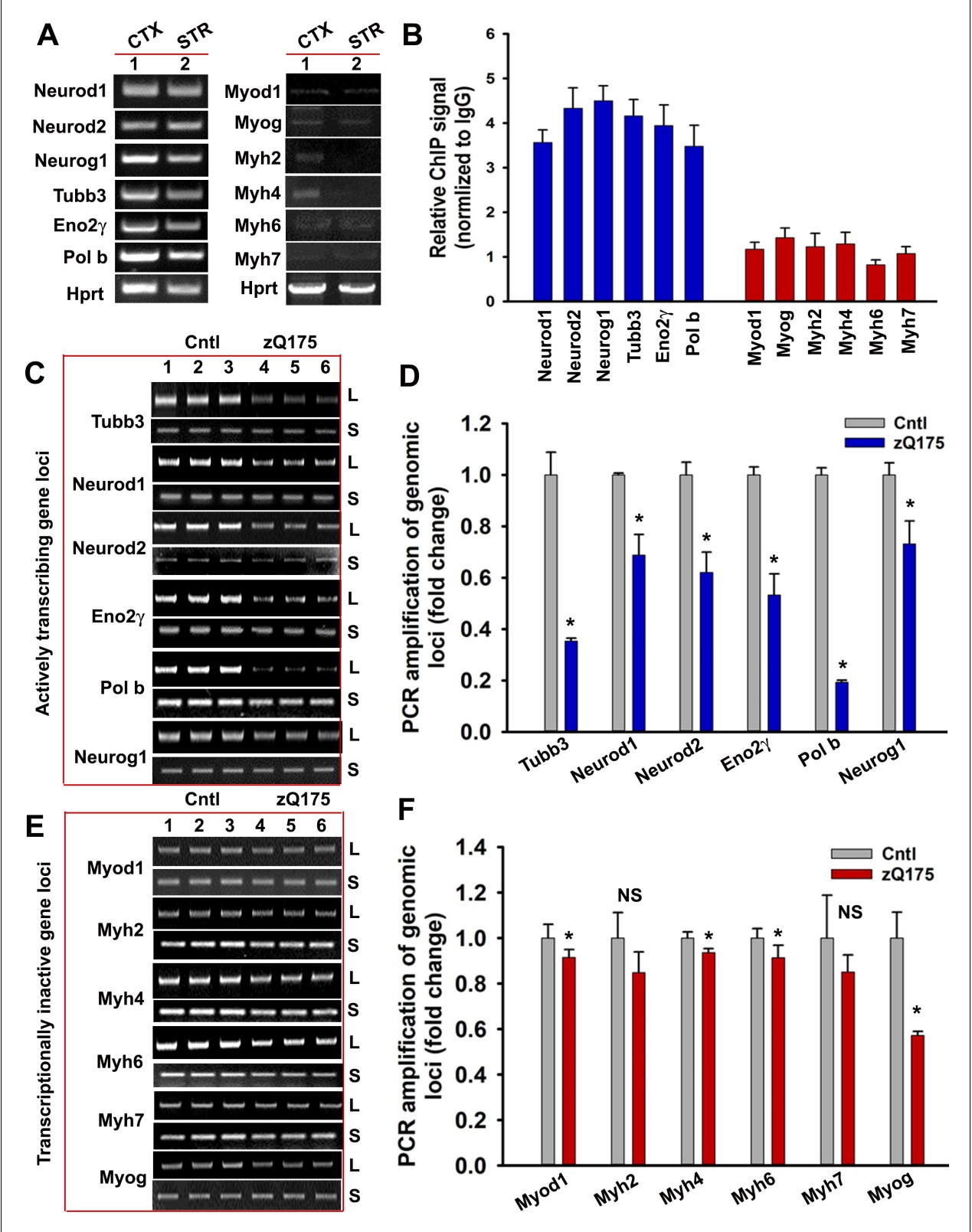

**Figure 6.** mHTT preferentially induces DNA damage/strand breaks in the transcriptionally active genome. (**A**) Tissue from CTX and STR of 5 WT mouse brain (7 weeks old) was pooled and total RNA was isolated and expression levels of various genes were measured using qRT-PCR analysis. Left panel shows amplified product of transcribing genes and the right panel for non-transcribing genes. (**B**) ChIP analysis showing relative occupancy of wtHTT on the actively transcribing (blue) vs. transcriptionally inactive (red) genome loci in 7 weeks old WT mouse STR. Three biological replicates and three

*Figure 6 continued on next page*

*Figure 6 continued*

technical replicates were used in this assay. Data represent mean ± SD. (**C**) Genomic DNA was isolated from the CTX of asymptomatic (7 weeks old) zQ175 transgenic (CTX from five transgenic mice were pooled) and age-matched WT control (CTX from 5 WT control mice were pooled) mice. Various transcriptionally active gene loci (Neurod1, Neurod2, Neurog1, Tubb3, Eno2γ, and Pol b) were PCR-amplified from the genomic DNA and analyzed on agarose gels; L: long amplicon (6 to 12 kb product), S: short amplicon (200–300 bp). (**D**) Relative PCR amplification efficacies of various actively transcribing gene loci in 7 weeks old WT control and zQ175 mouse brains (CTX). Data represent mean ± SD, *p<0.001. Five biological replicates each with three technical replicates were used in this assay. (**E**) PCR amplification of genomic DNA isolated from the CTX of asymptomatic (7 weeks) zQ175 and WT control mice and various loci that are transcriptionally inactive in brain (Myod1, Myog, Myh2, Myh4, Myh6, and Myh7) were PCR amplified. PCR products from WT control (lanes 1–3) and zQ175 mice (lanes 4 to 6) were analyzed on agarose gels. L: long amplicon (6 to 12 kb product), S: short amplicon (200–300 bp). (**F**) Relative amounts of PCR products from the transcriptionally inactive genomic loci in the CTX of WT control and zQ175 mice. Data represent mean ± SD; *p<0.001. Five biological replicates each with three technical replicates were used in this assay.

DOI: https://doi.org/10.7554/eLife.42988.021

The following source data and figure supplements are available for figure 6:

**Source data 1.** Mutant Huntingtin- mediated CBP degradation.
DOI: https://doi.org/10.7554/eLife.42988.026
**Figure supplement 1.** mHTT expression induces DNA strand breaks in STR but not in CRBL.
DOI: https://doi.org/10.7554/eLife.42988.022
**Figure supplement 2.** HD patients' brain and HD transgenic mouse brain accumulate DNA damages.
DOI: https://doi.org/10.7554/eLife.42988.023
**Figure supplement 3.** HD primary neurons accumulate DNA breaks preferentially in the actively transcribing genome.
DOI: https://doi.org/10.7554/eLife.42988.024
**Figure supplement 4.** N171-82Q transgenic mouse brain predominantly accumulates strand breaks in the transcriptionally active genome.
DOI: https://doi.org/10.7554/eLife.42988.025

substantial DNA damage accumulation in STR but negligible DNA damage accumulations in the CRBL (*Figure 6—figure supplement 1*). Moreover, immunostaining of the HD patients' brain and HD transgenic mouse brain sections with anti-phospho-53BP1 antibody, a DNA damage marker, showed increased presence of DNA damage as compared to control (*Figure 6—figure supplement 2*). Consistently, preferential accumulation of DNA strand breaks was observed in iPSC-derived HD primary neurons (Q50 and Q53) than controls (Q18 and Q28) (*Figure 6—figure supplement 3*). Increased DNA break accumulation was also observed in actively transcribing genes vs. non-transcribing genes in the N171-82Q transgenic CTX than the age-matched controls *Figure 6—figure supplement 4*). These data support our hypothesis that the HTT-TCR complex repairs strand breaks during transcription, and that this function is impaired by polyQ expansion, resulting in persistent strand break accumulation predominantly affecting actively transcribing genes in HD.

## HTT facilitates CBP degradation by inactivating ATXN3

Given that the deubiquitinase ATXN3 is present in the TCR complex, interacts with mHTT, and is sequestered in polyQ aggregates in HD brain, we postulated that compromised ATXN3 activity might increase ubiquitination and decrease levels of TCR components, adversely impacting complex functionality and transcription. To explore this possibility, we examined whether mHTT stimulates ubiquitination and degradation of specific TCR complex proteins. WB analyses of NEs from HD and control iPSC-derived primary neurons revealed a significant decrease in soluble CBP protein levels in HD neurons, whereas ATXN3, PNKP, POLR2A, and CREB levels were not affected (*Figure 7A & B*). Quantitative reverse transcription PCR analyses did not show a significant change in CBP mRNA levels upon mHTT expression (data not shown), suggesting that mHTT does not interfere with the expression of CBP in HD. This finding indicates that CBP might be degraded more in HD. However, an alternative possibility is that CBP becomes insoluble when post-translationally modified. Substantially reduced levels of CBP was also observed in the soluble fraction of proteins from cells expressing exogenous mHTT (data not shown). To determine whether abrogating ATXN3 activity causes reduced CBP levels, we measured TCR protein levels in ATXN3-depleted cells. Similar to HD iPSC-derived primary neurons, markedly lower CBP levels were observed in the soluble protein extract from the ATXN3-depleted cells (*Figure 7C & D*). Consistent with a previous report (*Cong et al., 2005*; *Jiang et al., 2003*), CBP levels were dramatically (~80%) reduced in the zQ175 CTX but only marginally (~20%) decreased in the CRBL (*Figure 7E & F*). To test whether ATXN3 interacts with

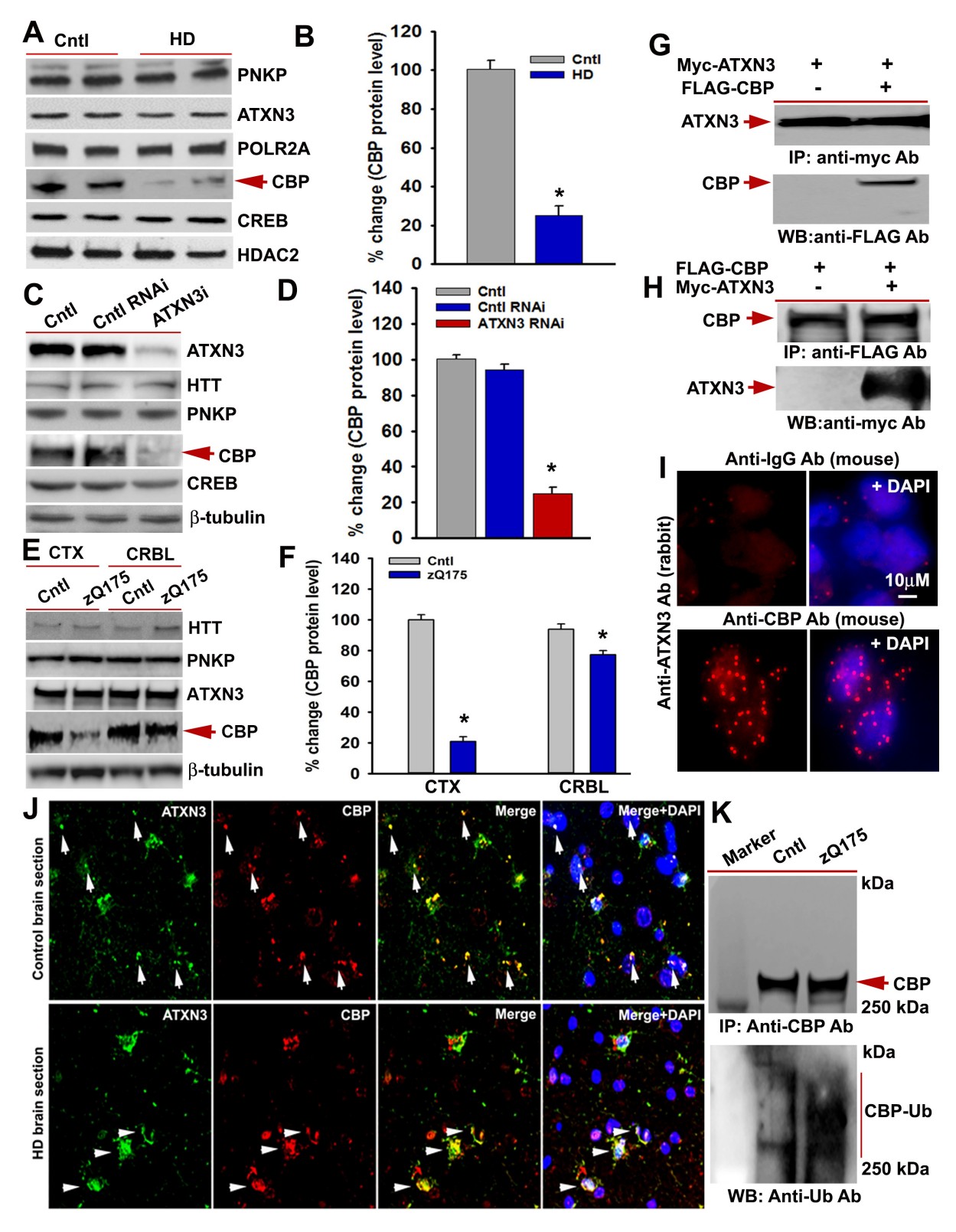

**Figure 7.** mHTT facilitates CBP degradation by inactivating ATXN3. (**A**) Nuclear extracts (NEs) isolated from control primary neurons (Q18 and Q28) and HD neurons (Q53 and Q109) were analyzed by WBs to measure PNKP, ATXN3, POLR2A, CBP, and CREB levels; HDAC2 was the loading control. (**B**) Relative CBP levels in control and HD neurons normalized to HDAC2. Two biological replicates and three technical replicates were used in this assay. Data represent mean ± SD; *p<0.001. (**C**) NEs isolated from SH-SY5Y cells expressing control shRNA or ATXN3 shRNA were subjected to WB to

*Figure 7 continued on next page*

*Figure 7 continued*

determine ATXN3, HTT, PNKP, CBP, and CREB levels; β-tubulin was used as loading control. (**D**) Relative CBP levels (normalized to β-actin) in WT control cells, cells expressing control RNAi or ATXN3 RNAi. Data represent mean ± SD; *p<0.001. (**E**) NEs isolated from CTX and CRBL of zQ175 and WT control mice, and analyzed by WB to determine HTT, PNKP, ATXN3, POLR2A and CBP levels; β-tubulin was the loading control. (**F**) Relative CBP levels in the CTX and CRBL in WT control and zQ175 mice. CBP levels were normalized to β-tubulin. Data represent mean ± SD; *p<0.001. (**G**) HEK293 cells were cotransfected with plasmids expressing Myc-ATXN3 and FLAG-CBP, Myc-ATXN3 IP'd with a Myc antibody, analyzed by WB to detect CBP in the Myc-IC (arrow). (**H**) HEK293 cells cotransfected with plasmids expressing Myc-ATXN3 and FLAG-CBP, NEs isolated and CBP IP'd with a FLAG antibody, ICs were subjected to WB to detect ATXN3 (arrow). (**I**) SH-SY5Y cells were analyzed by PLA to examine interactions between CBP and ATXN3. Nuclei were stained with DAPI (upper panel). Reconstitution of red fluorescence indicates interaction of CBP with ATXN3 (lower panel). (**J**) Control (upper panel) and HD (lower panel) patient brain sections were analyzed by immunostaining with antibodies against ATXN3 (green) and CBP (red) to assess their in vivo interactions. Merged red and green fluorescence appears as yellow/orange; nuclei stained with DAPI (blue). Arrow indicates the respective colocalization. (**K**) Total protein was isolated from control and zQ175 mice CTX, and CBP was IP'd with a CBP antibody (upper panel), and the IC analyzed with anti-ubiquitin antibody to detect CBP ubiquitination (lower panel).

DOI: https://doi.org/10.7554/eLife.42988.027

The following source data is available for figure 7:

**Source data 1.** DNA damage precedes neurodegeneration.
DOI: https://doi.org/10.7554/eLife.42988.028

CBP, we co-expressed Myc-ATXN3 and FLAG-CBP and IP'd Myc-ATXN3 from the NEs. WBs showed CBP in the Myc-IC (*Figure 7G*). Conversely, IP of the FLAG-CBP and subsequent WB revealed ATXN3 (*Figure 7H*). The PLA results also suggested intracellular interaction between ATXN3 and CBP (*Figure 7I*). Confocal microscopy showed distinct colocalization of ATXN3 and CBP in HD and control brain sections (*Figure 7J*, arrows). A recent study also showed significantly increased ubiquitination and reduced level of CBP in Hdh^{Q7/Q111} HD transgenic mouse brain (*Giralt et al., 2012*). To test whether mHTT expression increases CBP ubiquitination in zQ175 mouse brain, we IP'd CBP from the NE of zQ175 and control mouse brain. Consistent with a previous report (*Jiang et al., 2003*; *Giralt et al., 2012*), WBs of the CBP IC showed increased CBP ubiquitination in the transgenic brain (*Figure 7K*; lower panel). These data suggest that decreased ATXN3 activity due to its interaction with mHTT in the TCR complex may increase ubiquitination, and increased ubiquitination of CBP may cause aberrant localization of CBP, negatively impacting its solubility in HD. It is also possible that increased ubiquitination may facilitate degradation of CBP in HD.

## Discussion

Our findings reveal a critical proximal event by which polyQ expansions in mHTT induce DNA damage to activate the DDR ATM→p53 pro-apoptotic signaling cascade and disrupt tissue-specific transcriptional activity – key pathogenic features consistently described in HD (*Giuliano et al., 2003*; *Bae et al., 2005*; *Illuzzi et al., 2009*; *Bertoni et al., 2011*; *Xh et al., 2014*). A significant association of wtHTT with PNKP, ATXN3, POLR2A and associated transcription factors suggest that wtHTT may act as a scaffold factor to assemble various core components of the TCR complex. Our data suggest that formation of this functional TCR complex with wtHTT is essential for sensing and editing DNA lesions in the template strand during transcriptional elongation in post-mitotic differentiated neurons and may contribute in maintaining genome integrity and neuronal survival. Our results further indicate that interaction of PNKP with wtHTT stimulates its DNA end-processing activity to facilitate neuronal DNA repair. The role of wtHTT in maintaining TCR complex functionality and genome integrity is further validated by the fact that depletion of endogenous wtHTT protein dramatically depletes PNKP activity with a concurrent increase in DNA damage accumulation. In contrast, mHTT with polyQ expansions interacts with several key components of the complex but abrogates the activities of PNKP and ATXN3, thereby disrupting DNA repair and transcription leading to a possible early trigger for neurotoxicity and functional decline in HD (*Figure 8*).

Our data demonstrate that mHTT interaction with PNKP and the resultant decline in PNKP's enzymatic activity was evident in CTX and STR of HD transgenic mouse models but insignificant in the CRBL. Consistent with these findings, CTX and STR, the most affected brain regions in HD displayed extensive DNA strand breaks with impaired DNA repair capacity. In contrast, PNKP activity and genome integrity was marginally affected in CRBL, the brain region that is reported to be relatively

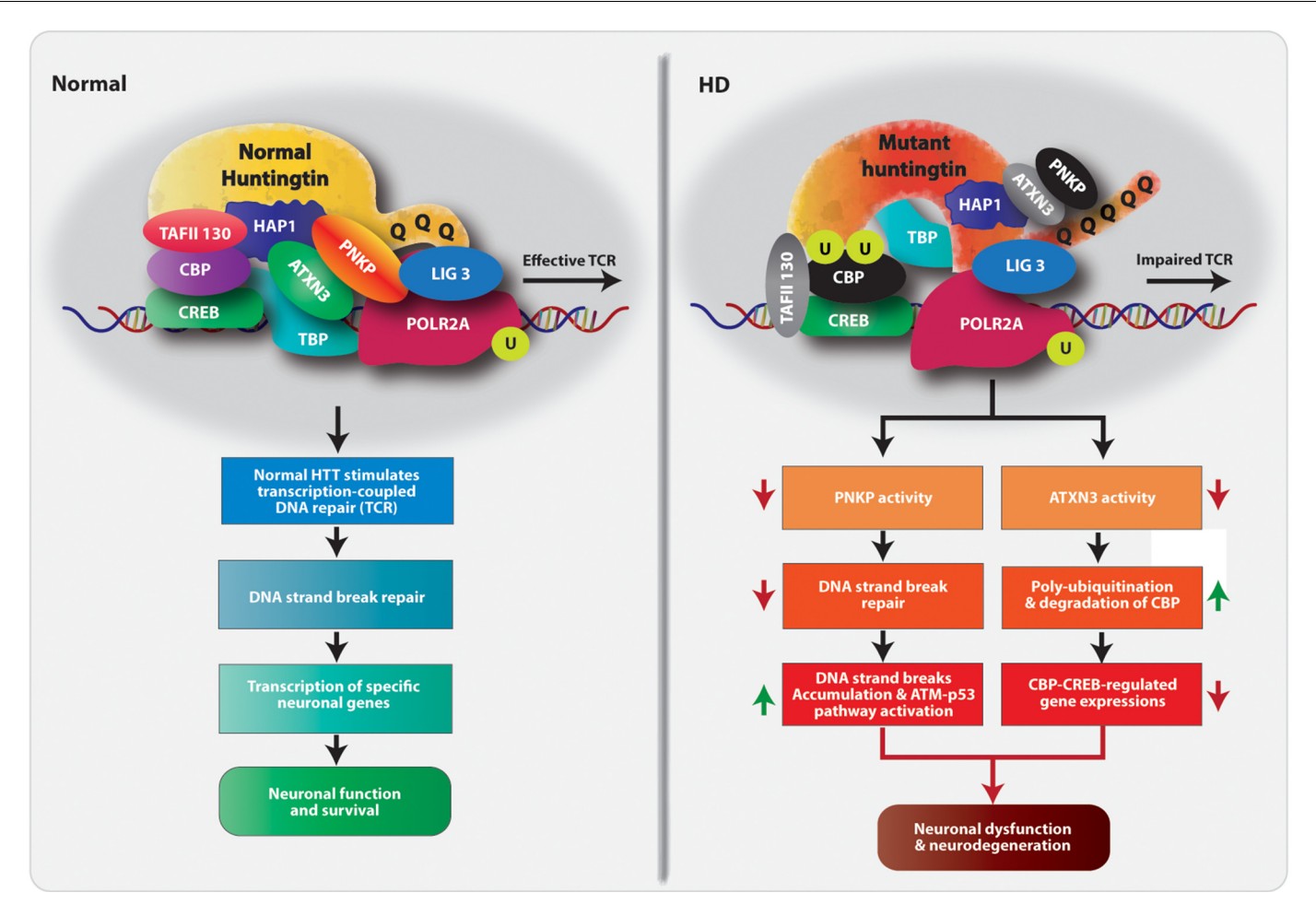

**Figure 8.** Proposed mechanism by which mHTT triggers neurotoxicity in HD. Schematic diagram of our hypothesized mechanism by which polyQ expansions in mHTT compromise the functional integrity of the TCR complex. Normal HTT forms a multiprotein TCR complex with POLR2A, ATXN3, PNKP, CBP, and additional DNA repair enzymes, and this structure monitors and edits DNA strand breaks/damage during transcriptional elongation, preserving genome integrity, transcription and neuronal survival. In HD, polyQ expansions in mHTT impair the normal function of the TCR complex; mHTT-mediated inactivation of PNKP activity impairs DNA repair, which leads to the persistence of DNA strand breaks and chronic activation of ATM-p53 pro-apoptotic signaling. Additionally, mHTT-mediated inactivation of ATXN3's deubiquitinating activity facilitates ubiquitination and degradation of CBP, impairing CBP-CREB-regulated gene transcription and further amplifying pro-degenerative output in the HD brain. PolyQ expansion in mHTT thus adversely impacts DNA repair and transcription and neural function and survival, triggering neurotoxicity and functional decline in HD.
DOI: https://doi.org/10.7554/eLife.42988.029

unaffected in HD. However, no alterations in the steady-state levels of PNKP, ATXN3 and other key TCR components were observed between CRBL vs. CTX or STR. Our present data do not explain why mHTT expression specifically impacts PNKP activity and genome integrity in the CTX and STR but spares CRBL. Further investigation is required to understand region-specific decreases in PNKP activity and DNA break accumulation in the HD brain. Complete characterization of the TCR complex in different brain regions may provide valuable insight into the selective neuronal vulnerability to mHTT-mediated toxicity. It will be interesting to understand the mechanism that imparts protection to CRBL against mHTT and could provide another node in the signaling pathway that could potentially be developed as a therapeutic target. Furthermore, more rigorous biophysical and structural studies are necessary with purified peptides/proteins to characterize the true nature of the protein-protein interactions and interacting partners to understand how this putative HTT-RNA polymerase complex maintains neuronal genome integrity and survival.

It is notable that the HTT-TCR complex preferentially associates only with the transcriptionally active genome both in vitro and in vivo. This suggests that the complex could be actively involved in repairing lesions in the template DNA strand during transcription and thus maintaining sequence integrity of the transcriptionally active genomes over the non-transcribing genome. This HTT-TCR complex may provide an additional layer of protective mechanism to maintain the sequence integrity of highly transcriptionally active genes in post-mitotic neurons. Depletion of PNKP activity by mHTT and subsequent accumulation of DNA strand breaks in the transcriptionally active genome extends our previous report (*Chakraborty et al., 2016*). Collectively, these data support our hypothesis that polyQ expansions in HTT result in the preferential accumulation of strand breaks in the transcriptionally active genome. Persistent DNA strand breaks in the actively transcribing genes may stall or impair transcription elongation, preventing adequate expression of a wide variety of neuronal genes and may contribute to the complexity and variability of HD pathology. Moreover, unrepaired lesions may further induce chronic activation of ATM→p53 signaling, as evidenced by increased phosphorylation of ATM, H2AX, and p53 in the HD brain (*Figure 5—figure supplement 1*). Chronic ATM-p53 pathway activation resulted in an increased expression of several of the p53 target genes may facilitate neuronal apoptosis in HD. These data support the hypothesis that decreased PNKP activity could be an important proximal event that triggers early neurotoxicity in HD; however, it remains to be tested whether restoration of PNKP activity and DNA repair efficiency can rescue genome integrity and structural and behavioral defects in HD models.

Our results provide evidence that ATXN3 is another key regulatory component of the TCR complex. Our data suggest that the mHTT-mediated decrease in ATXN3 activity either enhances degradation of specific TCR complex components or prevents appropriate formation of the TCR complex in HD. We propose that abrogating ATXN3 activity is a potential mechanism by which mHTT decreases CBP activity and thus adversely impacts the transcription of CREB-dependent genes in HD. ATXN3 binds and deubiquitinates both mono- and polyubiquitin chains in target proteins (*Burnett et al., 2003*; *Chai et al., 2004*). ATXN3 inactivation in mice increases protein ubiquitination (*Schmitt et al., 2007*), so diminished ATXN3 activity could influence CREB-regulated gene expression as described in HD (*Wyttenbach et al., 2001*). Since disruption of CREB activity in the brain triggers neurodegeneration (*Mantamadiotis et al., 2002*), mHTT-mediated decreases in ATXN3, CBP, and CREB activities might compromise neuronal function and trigger neurotoxicity, further amplifying pro-degenerative output in HD. The identification of HTT, CBP, PNKP, and ATXN3 as key regulatory components of the TCR complex and our description of how polyQ expansions disrupt the complex's functional integrity provide important insight into how mHTT could coordinately disrupt CREB-mediated transcription, increases DNA strand breaks, and activates ATM→p53 signaling. Collectively, these events compromise neuronal survival in HD. We hypothesize that POLR2A-mediated transcription might temporarily pause at DNA lesions, leading to mono-ubiquitination of specific TCR complex components, which signals complex assembly to stimulate and/or coordinate lesion repair in normal cells. ATXN3 deubiquitinates the components after repair, and normal transcription resumes. In contrast, the TCR complex stalls at strand breaks in mHTT-carrying cells, and due to reduced ATXN3 activity, specific complex component (s) are polyubiquitinated and accumulate aberrantly in polyQ inclusions (*Figure 8*). We propose that mHTT-mediated ATXN3 inactivation might impair CBP/CREB-dependent transcription, while reduced PNKP activity might result in DNA break accumulation and DDR pathway activation. The combination of chronic DDR signaling and dysregulation of CREB-dependent genes could trigger selective neuronal degeneration, a hallmark of HD. Defective DNA repair in post-mitotic neurons is an emerging causative factor of cognitive decline in neurodegenerative diseases (*Madabhushi et al., 2014*; *Madabhushi et al., 2015*; *Rass et al., 2007*). Consistent with our findings, point mutations in PNKP result in microcephaly and seizures (*Shen et al., 2010*), whereas a frame-shift mutation in the PNKP gene was identified in a neurodegenerative disorder characterized by epilepsy (*Poulton et al., 2013*). Therefore, mHTT-mediated ablation of PNKP activity could lead to impaired DNA repair, persistent accumulation of DNA strand breaks that may in part contribute to neurotoxicity and neuronal dysfunction in HD.

This study provides multiple lines of evidence suggesting that mHTT-mediated loss of DNA repair and deubiquitinating activity could possibly be critical proximal events that impair the TCR. This could provide a mechanistic link between transcriptional dysregulation leading to aberrant activation of ATM-dependent pro-degenerative pathways and early neurotoxicity in HD. Although the final biological output triggered by impaired TCR and unrepaired DNA strand breaks in HD remains to be

fully described, the present data indicate a potential mechanism by which polyQ expansions in mHTT could disrupt the functional integrity of TCR complex and compromises transcriptional regulation and genomic integrity in post-mitotic neurons. Molecular strategies that interfere with the interaction of mHTT with the TCR complex could reduce neurotoxicity and slow functional decline in HD. Alternatively, molecular approaches to stimulate PNKP activity could be a reasonable way to combat transcriptional dysregulation and inappropriate activation of pro-apoptotic signaling in HD. Our findings could help elucidate the cell type-specific pattern of pathology in HD. We propose the possibility that the compromised TCR efficiency in the basal ganglia or cortex could render these neuronal populations more vulnerable. Collectively, our findings suggest an intriguing molecular mechanism that could explain how mHTT expression in HD could compromise genome integrity and neuronal survival.

## Materials and methods

### Plasmid construction

The construction of plasmids expressing the N-terminal fragment of HTT (exon1: NT-HTT-Q23 and NT-HTT-Q148) and full-length HTT (FL-HTT-Q23 and FL-HTT-Q148) was described previously (*Tanaka et al., 2006*). The N-terminal fragments of wtHTT and mHTT were sub-cloned in pAcGFPC1 (Clontech, USA) to construct pGFP-NT-HTT-Q23 and pGFP-NT-HTT-Q97, respectively. The number of CAG repeats contracted to 97 after propagation in *Escherichia coli*. The plasmids pGFP-NT-HTT-Q23 and pGFP-NT-HTT-Q97 were digested with NheI and MluI, and the fragments containing GFP-NT-HTT-Q23 and GFP-NT-HTT-Q97 were sub-cloned into the TET-inducible responder plasmid pTRE3G (Clontech, USA) using appropriate linkers. The plasmid pTet-ON (Clontech, USA) and responder plasmids (pTRE-GFP-NT-HTT-Q97 or pTRE-GFP-NT-HTT-Q23) were transfected into SH-SY5Y cells, and clones were selected with neomycin. The stable inducible clones expressing GFP-NT-HTT-Q97 or GFP-NT-HTT-Q23 were incubated with medium containing doxycycline (500 ng/mL), and transgene expression was assessed by WB using anti-GFP antibodies. The PNKP cDNA was cloned into pcDNA3.1/hygro (Invitrogen, USA) to construct pRPS-PNKP, which was transfected into SH-SY5Y cells encoding inducible GFP-NT-HTT-Q23 and GFP-NT-HTT-Q97. The clones were selected for hygromycin resistance. PNKP expression was examined by WB, and PNKP activity was assessed as described previously (*Chatterjee et al., 2015*). To express PNKP and its functional domains as FLAG-tagged peptides, the full-length cDNA and FHA domain (1–300 amino acids), kinase domain (131–337 amino acids), phosphatase domain (338–521 amino acids), FHA and kinase domain (1–337 amino acids), and kinase and phosphatase domain (131–521 amino acids) were PCR-amplified using specific primers and cloned into plasmid pCMV-DYKDDDDK (Clontech, USA).

### Plasmids for the bimolecular fluorescence complementation assay

Plasmids pBiFC-VN173 (encoding 1 to 172 N-terminal amino acids of cyan fluorescent protein, CFP) and pBIFC-VC155 (encoding 155 to 238 C-terminal amino acids of CFP) were kindly provided by Dr. Chang-Deng Hu (Addgene plasmids 22011 and 22010). The N-terminal fragments of HTT cDNA (encoding 23 or 97 glutamines) were cloned in-frame with the C-terminal amino acids of CFP in plasmid pBiFC-VC155 to construct pVC-NT-HTT-Q23 and pVC-NT-HTT-Q97, respectively. Full-length PNKP or its catalytic domain (phosphatase and kinase domains, 131–521 amino acids) was cloned in plasmid pBIFC-VN173 to construct pVN-PNKP or pVN-(PHOS + KIN)-PNKP, respectively. SH-SY5Y cells (2 × 10^5 cells) were grown on chamber slides and transfected 24 hr later. Plasmids pVN-(PHOS + KIN)-PNKP and pVC-HTT-Q23 or pVN-(PHOS + KIN)-PNKP and pVC-HTT-Q97 were cotransfected, and reconstitution of the green/yellow fluorescence of CFP was monitored by fluorescence microscopy.

### Cell culture and plasmid transfection

Human neuroblastoma SH-SY5Y cells were purchased from ATCC (Cat # CRL-2266) and cultured in Dulbecco's minimum essential medium (DMEM) containing 15% fetal bovine serum (FBS), and 1% B-27 (Invitrogen, USA). SH-SY5Y cells stably encoding inducible GFP-NT-HTT-Q23 or GFP-NT-HTT-Q97 were cultured in DMEM, and transgene expression was induced by adding doxycycline to the medium to a final concentration of 500 ng/mL. PC12 cells carrying full-length wtHTT-Q23 or mHTT-

Q148 were cultured in DMEM containing 15% FBS and doxycycline (500 ng/mL). HTT expression was induced by withdrawing doxycycline from the media for 5–7 days, and transgene expression was verified by WB. Plasmids expressing the RNAi targeting ATXN3 were from Dharmacon, USA. SH-SY5Y cells were transfected with the ATXN3-RNAi plasmids using Lipofectamine RNAi-MAX reagent (Invitrogen, USA); stable cells were selected for puromycin resistance and differentiated in DMEM containing 5 μM retinoic acid. All the cell lines were authenticated by short tandem repeat analysis in the UTMB Molecular Genomics Core. The possible mycoplasma contaminations in all the cell lines were tested using GeM Mycoplasma Detection Kit (SIGMA, Cat# MP0025) using a PCR based screening method and cells were found to be free from mycoplasma contamination.

## Analysis of HTT-associated TCR proteins by co-immunoprecipitation (Co-IP)

Co-IP from NEs: NEs from SH-SY5Y cells were isolated and treated with benzonase to remove DNA and RNA to avoid nucleic acid-mediated Co-IP. Specific target proteins were IP'd, and the IC was washed extensively with cold Tris-buffered saline (50 mM Tris-HCl [pH 7.5] 200 mM NaCl) containing 1 mM EDTA, 1% Triton-X100, and 10% glycerol. The complexes were eluted from the beads with 25 mM Tris-HCl (pH 7.5) and 500 mM NaCl and analyzed by WB.

Co-IP from tissue: Approximately 250 mg of cortex from freshly sacrificed WT mice was harvested and homogenized with 4 volumes of ice-cold buffer (0.25 M sucrose, 15 mM Tris-HCl [pH 7.9], 60 mM KCl, 15 mM NaCl, 5 mM EDTA, 1 mM EGTA, 0.15 mM spermine, 0.5 mM spermidine, 1 mM dithiothreitol, 0.1 mM phenylmethylsulfonyl fluoride [PMSF], and protease inhibitors [Roche Applied Science, Germany]) with ~20 strokes to disrupt tissues (*Chakraborty et al., 2015*). Homogenization was continued until a single-cell slurry was obtained, incubated on ice for 15 min, and centrifuged at 1,000 × g to obtain the cell pellet. NEs were then prepared from the cell pellet for co-IP analysis. The ICs were analyzed by WB to identify interacting protein partners.

## HD autopsy brain tissue samples

Human autopsy specimens were obtained in accordance with local legislation and ethical rules. Control brain samples were collected from age-matched individuals without neurodegenerative disorders. The HD brain tissue samples were obtained from patients with HD who were clinically characterized based on the presence of chorea and motor, mood, and cognitive impairment. The molecular diagnosis of HD was established by analyzing genomic DNA extracted from peripheral blood using a combination of PCR and Southern blotting. HTT CAG repeat lengths were established by sequencing the expansion loci of the mutant allele. All brain autopsies were immediately frozen in liquid nitrogen and stored at −80°C until further analysis.

## HD iPSC differentiation

Products purchased from Thermo Fisher Scientific (Waltham, MA USA), unless otherwise specified. Three control: CS25iCTR18n6, CS14iCTR28n6, CS83iCTR33n1 and three HD: CS87iHD50n7, CS03iHD53n3 and CS09iHD109n1 iPSC lines were derived and cultured as previously described on hESC-qualified Matrigel (HD iPSC *HD iPSC Consortium, 2017*). Once at 70% confluency, neural induction and further differentiation of neural progenitors with the addition of Activin A (Peprotech, USA), was performed as previously described in *Telezhkin et al. (2016)*. Neuronal maturation was performed as previously described (*Telezhkin et al., 2016*) on Nunc six well plates. After 3 weeks of maturation, medium was removed and cells were washed once with PBS pH 7.4, without $Mg^{2+}$ and $Ca^{2+}$. Subsequently, cells were washed with 4°C PBS pH 7.4, without $Mg^{2+}$ and $Ca^{2+}$, and scraped using a cell scraper, pipetted into a centrifuge tube and centrifuged at 250 x g for 3 min. PBS was removed and samples were flash frozen in liquid nitrogen.

## HD transgenic mice

The HD knock-in mouse model zQ175 expresses full-length mHTT from the endogenous mouse *Htt* promoter (*Menalled et al., 2012*). The N171-82Q transgenic mouse line expresses the truncated N-terminus of human HTT cDNA with a polyQ repeat length of 82 under control of the mouse prion promoter (*Schilling et al., 1999*). Heterozygous transgenic mice and control non-transgenic littermates (n = 4–5 pools of two animals per genotype) were sacrificed, and fresh brain tissues were

used for enzyme assays, isolating genomic DNA, and obtaining protein for WB analyses. For immunofluorescence assays, transgenic and control littermate mice were deeply anesthetized and transcardially perfused with sterile phosphate-buffered saline (PBS) followed by 4% paraformaldehyde in PBS. Brains were post-fixed overnight in fixative solution and embedded in OCT and stored in liquid nitrogen. Slides with 4-μm-thick frozen sections were processed for immunostaining with appropriate antibodies. All procedures involving animals were in accordance with the National Institutes of Health Guide for the care and use of Laboratory Animals, and approved by the Institutional Animal Care and Use Committee of University of California Irivine (protocol #: AUP-18–155); and Duke University (protocol #: A225-17-09).

## Alkaline comet assays

Alkaline comet assays were performed using a Comet Assay Kit (Trevigen, USA). Cells were suspended in 85 μL ice-cold PBS and gently mixed with an equal volume of 1% low-melting-point agarose. The cell suspension was dropped onto an agarose layer and incubated in lysis buffer for 1 hr. After lysis, slides were incubated in buffer containing 0.3 M NaOH, 1 mM EDTA (pH 13) for 40 min and electrophoresed for 1 hr. After neutralization, slides were stained and analyzed with a fluorescence microscope.

## Antibodies and WB analysis

Cell pellets or brain tissues were homogenized, and total protein was isolated using a protein extraction kit (Millipore, USA). The cytosolic and nuclear fractions were isolated from cells/tissue using a NE-PER protein extraction kit (Thermo Scientific, USA). WBs were performed according to the standard procedure, and each experiment was performed at least three times to ensure statistically significant results. The antibodies for p53 (Cat #9282), p53-S15 (Cat #9286), p53-S20 (Cat #9287), p53-S46 (Cat #2521), Chk2 (Cat #2662), Chk2-T68 (Cat #2661), CBP (Cat #7389) and APE1 (Cat #4128) were from Cell Signaling, USA; anti-H2AX (Cat #ab11175) and γH2AX-S139 (Cat #ab11174) were from Abcam, UK; anti-ATM (Cat #1549–1) and ATM-S1981 (Cat #2152–1) were from Epitomics, USA, or anti-ATM from Santa Cruz (sc-23921), anti-ataxin-3 monoclonal antibody (Cat #MAB 5360), monoclonal anti-HTT antibody (MAB 2170) and 5TF1-1C2 (Mab1574) were from Millipore, USA. Rabbit polyclonal HAP-1 (Cat #TA306425) was from Origene, USA, and mouse monoclonal HAP-1 (MA1-46412) was from Thermo Scientific, USA. RNA pol II (sc-899) and DNA ligase 3 (sc-135883) were from Santa Cruz Biotechnology, USA. PNKP rabbit polyclonal antibody (Cat #MBP-1-A7257) was from Novus Biologicals, USA, and BioBharati Life Science (Cat# BB-AB0105), India, and PNKP mouse monoclonal antibody was a kind gift from Dr. Michael Weinfeld (University of Alberta, Canada).

## Immunohistochemical analysis

SH-SY5Y cells or frozen brain sections were immunostained with anti-PNKP, HTT, CBP, POLR2A, ATXN3, and anti-polyQ 5T1-1C2 antibodies. Nuclei were stained with DAPI (Molecular Probe, USA) and imaged under a confocal microscope.

## Cell toxicity assay

Expression of mHTT or wtHTT was induced in PC12 cells by removing doxycycline from the culture medium for 4–7 days. Induced cells were dissociated with Accutase (Gibco), and collected by centrifugation. Cell toxicity was assayed using a commercially available Annexin-V Cell Toxicity Assay kit (4830–01 K, Trevigen, USA). $1 \times 10^6$ Cells were incubated at room temperature with 1 μl Annexin-V-FITC (1 μg/ml) and 5 ul Propidium Iodide, in the provided binding buffer, for 15 min, before diluting with binding buffer. FITC fluorescence was analyzed by flow cytometry using a Cytoflex (Beckman Coulter), measuring 10,000 events per sample. Gating on main cell population was performed by FSC/SSC gating. Positive thresholds determined with unstained negative control, and $H_2O_2$ treated positive control samples. Identical thresholds applied to all samples. Data were analyzed using CytExpert software (Beckman Coulter).

## Image collection

Images were collected using a Zeiss LSM-510 META confocal microscope with 40 × or 60 × 1.2 numerical aperture water immersion objectives. Images were obtained using two excitation

wavelengths (488 and 543 nm) by sequential acquisition. Images were collected using 4-frame-Kall-man-averaging with a pixel time of 1.26 µs, a pixel size of 110 nm, and optical slices of 1.0 µm. Z-stack acquisition was performed at 0.8 µm steps. Orthogonal views were processed with LSM 510 software.

## Caspase-3 activity measurements

Caspase-3 activities were measured using a Caspase-3 assay kit (BD Biosciences, USA) based on hydrolysis of the substrate acetyl-Asp-Glu-Val-Asp p-nitroanilide (Ac-DEVD-pNA), resulting in release of the p-nitroaniline (pNA) moiety. Released pNA is detected at 405 nm. Comparison of pNA absorbances from the sample and control allows determination of the fold increase in caspase-3 activity (relative caspase-3 activity is expressed in arbitrary units).

## In situ proximity ligation assay (PLA)

SH-SY5Y cells were plated on chamber slides and cultured in DMEM for 24 hr. SH-SY5Y cells or brain sections were fixed with 4% paraformaldehyde, permeabilized with 0.2% Tween-20, washed with 1 × PBS, incubated with primary antibodies for PNKP (mouse monoclonal), HTT (rabbit polyclonal and mouse monoclonal), PNKP (mouse monoclonal), POLR2A (rabbit polyclonal), HAP-1 (rabbit polyclonal and mouse monoclonal), ATXN3 (rabbit polyclonal and mouse monoclonal), and DNA ligase 3 (rabbit polyclonal). These samples were subjected to PLAs using the Duolink PLA kit (O-Link Biosciences, Sweden). Nuclei were stained with DAPI, and PLA signals were visualized under a fluorescence microscope at 20 × magnification.

## PNKP activity measurements

The 3'-phosphatase activity of PNKP in the nuclear extract (250–500 ng) of cells/mouse brains or with purified recombinant His-tagged PNKP (25 fmol) was conducted as we described previously (*Wiederhold et al., 2004*; *Mandal et al., 2012*; *Chatterjee et al., 2015*). Nuclear extracts for the 3' phosphatase assay was prepared following standard protocols from cells (*Chakraborty et al., 2016*) or mouse brains tissues (*Chakraborty et al., 2015*). A $^{32}$P-labeled 3'-phosphate-containing 51-mer oligo substrate with a strand break in the middle (5 pmol) was incubated at 37°C for 15 min in buffer A (25 mM Tris-HCl, pH 7.5, 100 mM NaCl, 5 mM MgCl$_2$, 1 mM DTT, 10% glycerol and 0.1 µg/µl acetylated BSA) with 5 pmol of unlabeled (cold) substrate. The reaction was stopped by adding buffer B (80% formamide, 10 mM NaOH) and the reaction products were electrophoresed on a 20% Urea-PAGE to measure the amount of 3' phosphate release from the radio-labeled substrate. The radioactive bands were visualized in PhosphorImager (GE Healthcare, USA). The data were represented as % of the phosphate release (% product) with the total radiolabeled substrate as 100.

## Total DNA repair assay

Total DNA repair assays were carried out according to the protocol of *Wiederhold et al. (2004)*. Briefly, 10 pmol DNA substrate (a 51-mer DNA-oligo) annealed to two shorter DNA duplexes, one containing 3'-P and the other with 5'-P with a 4-nt gap in the middle was used to assess total repair activity (DNA end cleaning +gap filling through polymerization +ligation to fill the ends) in NEs (2.5 µg) from wtHTT- and mHTT-expressing neuronal cells and zQ175 and control mouse brain samples. Total repair activity was also assessed with the same substrate with DNA oligos containing 3'-OH (clean DNA ends). In both cases, the 20 µL reaction mixture contained 1 mM ATP, 50 µM unlabeled dNTPs, and 0.5 pmol [α−32P]-dCTP (the concentration of cold dCTP was lowered to 5 µM) in BER buffer and incubated for 45 min at 30°C. The reaction products were analyzed with 20% urea-polyacrylamide gel electrophoresis, and the radioactive bands were detected in a Phosphorimager (GE Life Sciences, USA).

## Gene expression analysis by real-time quantitative RT-PCR

Freshly dissected brain tissue from transgenic and age-matched control mice was homogenized in TRIzol (Thermo Scientific, USA), and total RNA was extracted using an RNA extraction kit (Qiagen, USA) and purified using a DNA-free DNAse Kit (Ambion, USA). Next, 1 µg of total RNA was reverse-transcribed using an RT-PCR kit (Clontech, USA). A cDNA aliquot from each reaction was quantified,

and 500 ng of cDNA from each reaction was used for qRT-PCR. 18S rRNA was used as control for the qRT-PCR analysis. The reactions were repeated three times using the following primers.

Neurod1: F: AGCCCTGATCTGGTCTCCTT; R: CTGGTGCAGTCAGTTAGGGG
Neurod2: F: AAGCCAGTGTCTCTTCGTGG; R: TTGGACAGCTTCTGCGTCTT
Neurog1: F: CCAGGACGAAGAGCAGGAAC; R: GGTCAGAGAGTGGTGATGCC
Tubb3: F: TGAGGCCTCCTCTCACAAGT; R: ACCACGCTGAAGGTGTTCAT
Eno2 γ: F: CCCAGGATGGGGATTTTGCT; R: CCTCCCCTGATCTGCTACCT
Pol b: F: TTCCACCGGTAAGACCCAGG; R: GCCAGTAACTCGAGTCAGGA
Myod1: F: AGCATAGTGGAGCGCATCTC; R: TTGGGGCTGGATCTAGGACA
Myog: F: GAGGAAGTCTGTGTCGGTGG; R: CCACGATGGACGTAAGGGAG
Myh2: F: CGAGAGACGAGTGAAGGAGC; R: GAATCACACAGGCGCATGAC
Myh4: F: AGCGCAGAGTGAAGGAACTC; R: TCTCCTGTCACCTCTCAACAGA
Myh6: F: ATAAAGGGGCTGGAGCACTG; R: TCGAACTTGGGTGGGTTCTG
Myh7: F: CCTTACTTGCTACCCTCAGGTG; R: GGCCATGTCCTCGATCTTGT
Gapdh F: ATGAGAGAGGCCCAGCTACT; R: TTTGCCGTGAGTGGAGTCAT
Bcl2L11: F: TTGGATTCACACCACCTCCG; R: CGGGATTACCTTGCGGTTCT
Pmaip 1: F: CTCGCTTGCTTTTGGTTCCC; R: ACGACTGCCCCCATACAATG
Bid: F: CCACAACATTGCCAGACATCTCG: R: TCACCTCATCAAGGGCTTTGGC
Pidd: F: ACAGAAGAGCCTCGGCAAGTCT: R: GAAAGGCACAGCAGAGGGCTTA
Apaf1: F: CACGAGTTCGTGGCATATAGGC: R: GGAAATGGCTGTCGTCCAAGGA

## Chromatin immunoprecipitation (ChIP)

ChIP assays were performed using fresh brain tissue of WT mice as previously described (*Chakraborty et al., 2016*; *Sailaja et al., 2012*). Briefly, 80–100 mg of freshly harvested CTX was chopped into small pieces and fixed in 1% formaldehyde for 15 min. The samples were centrifuged at 440 × g for 5 min at room temperature, and 0.125 M glycine was added to terminate cross-linking. The samples were washed two to three times with ice-cold PBS (containing protease inhibitors) and centrifuged each time at 440 × g for 4 min at 4°C. The pellet was resuspended in 1 mL ice-cold lysis buffer (10 mM EDTA, 1% [w/v] SDS, 50 mM Tris-HCl [pH 7.5]) with protease inhibitors and PMSF for 15 min and homogenized to produce a single-cell suspension. The samples were then transferred to pre-cooled 1.5 mL tubes and centrifuged at 2260 × g for 5 min. The pellet was resuspended in lysis buffer and sonicated to generate ~500 bp DNA fragments. The samples were centrifuged at 20,780 × g for 30 min at 4°C, and supernatants were collected for ChIP. The sheared chromatin was IP'd for 6 hr at 4°C with 10 µg isotype control IgG (Santa Cruz Biotechnology, USA: sc-2027) or anti-HTT antibody. After DNA recovery with proteinase K treatment followed by phenol extraction and ethanol precipitation, 1% of input chromatin and the precipitated DNA were analyzed by qPCR with the following primers. ChIP data are presented as percent binding relative to the input value.

**Neurod1:**
F: CTGCAAAGGTTTGTCCCAAGC; R: CTGGTGCAGTCAGTTAGGGG
**Neurod2:**
F: CAGGCCCTCCCAAGAGACTT; R: TCGTGTTAGGGTGAAGGCGT
**Neurog1:**
F: GCTTGCTCCAGGAAGAACCT; R: AGAGACACCGCTACTAGGCA
**Tubb3:**
F: GTGGGGCTCTCCCCTAAAAC; R: TTGGGAGCGCACAGTTAGAG
**Eno2 γ:**
F: TAGGGGTGCCTAGTCCTGTC; R: GAGTGCTGGATGTGTGGTCA
**Myod1:**
F: ATCTGACACTGGAGTCGCTTT; R: TTAGTCTCAGCTGCTGGTTCC
**Myog:**
F: GGCCACCAGAGCTAGAACAG; R: ATGAAGGCTGTGGACTTGGG
**Myh2:**
F: TCAGTGAGCAGTGGGAGCTA; R: GTACAAACACGGGGACACCC
**Myh4:**
F: AGGTGTACAACTCCGTGGGT; R: GCTCTAGCAAGACCAGTCACG

**Myh6:**
F: TCGTGCCTGATGACAAGGAG; R: CTTTCTGGCAAGCGAGCATC
**Myh7:**
F: ATTGGTGCCAAGGTGGGTTT; R: CCTGGGGTTCCCAGAATCAC

## LA-qPCR analysis to assess DNA strand breaks

LA-qPCR assays were carried out following an existing protocol (*Santos et al., 2006*). Briefly, tissues were harvested from the cortex (CTX), striatum (STR), and cerebellum (CRBL) of control and HD transgenic mice, and genomic DNA was extracted using the genomic-tip 20/G kit (Qiagen, Germany). Genomic DNA was quantified, and gene-specific LA-qPCR analyses were performed using Long Amp Taq DNA polymerase (NEB, USA). Various genomic loci were PCR-amplified from actively transcribing genes in brain (e.g., neuronal differentiation factor 1 and 2 [Neurod1 and Neurod2], neurogenic basic-helix-loop-helix protein neurogenin 1 [Neurog1], tubulin beta three class III [Tubb3], neuron-specific enolase 2 [Eno2γ], and DNA polymerase β [Pol b]). Non-transcribing loci (e. g., myogenic differentiation factor 1 [Myod1]; myogenic factor 4; myogenin [Myog]; and myosin heavy chain 2, 4, 6, or 7 [Myh2, Myh4, Myh6, or Myh7]) were amplified using the primers listed below. Loci from genomic DNA isolated from iPSC-derived control and HD primary neurons were PCR-amplified with the primers listed below. The cycle numbers and DNA concentrations were standardized before each final reaction so that the reaction remained within the linear amplification range (*Santos et al., 2006*). The final PCR conditions were optimized at 94°C for 30 s (94°C for 30 s, 55–60°C for 30 s depending on the oligo annealing temperature, 65°C for 10 min) for 25 cycles and 65°C for 10 min. Each reaction used 15 ng of DNA template, and the LA-qPCRs for all studied genes used the same stock of diluted DNA samples to avoid amplification variations due to sample preparation. A small DNA fragment for each gene was amplified to normalize large fragment amplification. The PCR conditions were 94°C for 30 s, 54°C for 20 s, 68°C for 30 s for 25 cycles, and 68°C for 5 min. Short PCR used 15 ng of the template from the same DNA aliquot. The amplified products were visualized on gels and quantified with the ImageJ software based on three independent replicate PCRs. The extent of damage was calculated according to our previously described method (*Chakraborty et al., 2016*).

## Mouse neurod1

Long: F: CTCGCAGGTGCAATATGAATC; R: GCAACTGCATGGGAGTTTTCT
  Short: F: CTGCAAAGGTTTGTCCCAAGC; R: CTGGTGCAGTCAGTTAGGGG

## Mouse neurod2

Long: F: GGCAGTGGTTGGGATGGTAT; R: CTCACTCTGTGCTGTCTGTCTC
  Short: FP: CAGGCCCTCCCAAGAGACTT; R: TCGTGTTAGGGTGAAGGCGT

## Mouse neurog1

Long: F: GATGAGCCCCTGAAGACGAG; R: GCCAATCTTGCTTCTTGCGT
  Short: F: GCTTGCTCCAGGAAGAACCT; R: AGAGACACCGCTACTAGGCA

## Mouse tubb3

Long: F: GGTACAGGGGATGTGGTTGG; R: GAGTCTCCTGCCTGTCCCTA
  Short: F: GTGGGGCTCTCCCCTAAAAC; R: TTGGGAGCGCACAGTTAGAG

## Mouse eno2 γ

Long: F: CTTGTTCTTCGGGGACCCTC; R: CATCCGTGTGCTTAAGGGGT
  Short: F: TAGGGGTGCCTAGTCCTGTC; R: GAGTGCTGGATGTGTGGTCA

## Mouse pol B

Long: F: TATCTCTCTTCCTCTTCACTT; R: GTGATGCCGCCGTTGAGGGTCTCCTG
  Short: F: TATGGACCCCCATGAGGAACA; R: AACCGTCGGCTAAAGACGTG

### Mouse myod1

Long: F: ATAGACTTGACAGGCCCCGA; R: GGACCGTTTCACCTGCATTG
  Short: F: ATCTGACACTGGAGTCGCTTT; R: TTAGTCTCAGCTGCTGGTTCC

### Mouse myog

Long: F: ACAAGCCTTTTCCGACCTGA; R: CCATGGCCAAGGCGACTTAT
  Short: F: GGCCACCAGAGCTAGAACAG; R: ATGAAGGCTGTGGACTTGGG

### Mouse myh2

Long: F: ATCTCAGGAGCACCCATCCT; R: GAAAAGGGTGTGCCAAGCAG
  Short: F: TCAGTGAGCAGTGGGAGCTA; R: GTACAAACACGGGGACACCC

### Mouse myh4

Long: F: GACGTGGAACTGTTAGGCCA; R: AAGCCAGAGTCTTCAACCCG
  Short: F: AGGTGTACAACTCCGTGGGT; R: GCTCTAGCAAGACCAGTCACG

### Mouse myh6

Long: F: GACAAGGGGCATTGTAGCCT; R: TCTGCCTACCTTATGGGGCT
  Short: F: TCGTGCCTGATGACAAGGAG; R: CTTTCTGGCAAGCGAGCATC

### Mouse myh7

Long: F: TTTGGGTTGGCCTGTCAGTT; R: ATCCCTAGCTGGGGCTTGTA
  Short: F: ATTGGTGCCAAGGTGGGTTT; R: CCTGGGGTTCCCAGAATCAC

### Human TUBB3

Long: F: TGCTTCTCATGCTTGCTACCAC; R: TCTGTCCCTGTAGGAGGATGT
  Short: F: CCTGTCCCTTTGTTGGAGGG; R: CGAGGTGGGCTAACAATGGA

### Human NEUROD1

Long: F: CCGCGCTTAGCATCACTAAC; R: TGGCACTGGTTCTGTGGTATT
  Short: F: TGCCTCTCCCTTGTTGAATGTAG; R: TTCTTTTTGGGGCCGCGTCT

### Human POLB

Long: F: CATGTCACCACTGGACTCTGCAC R: CCTGGAGTAGGAACAAAAATTGCT
  Short: F: AGTGGGCTGGATGTAACCTG R: CCAGTAGATGTGCTGCCAGA

### Human ENO2 γ

Long: F: ACGTGTGCTGCAAGCAATTT; R: CCTGAAACTCCCCTGACACC
  Short: F: GGTGAGCAATAAGCCAGCCT; R: CAGCTTGTTGCCAGCATGAG

### Statistical analysis

Data reported as mean ± SD and the statistical analysis was performed using Sigma Plot (SYSTAT Software). Differences between two experimental groups were analyzed by Student's t test (2-tail, assuming unequal variances). When comparing multiple groups, One-way ANOVA was performed followed by Tukey's post-hoc test to determine significance. In all cases, $p < 0.05$ was considered significant.

## Acknowledgments

This research was supported by a Mitchell Center Grant (University of Texas Medical Branch, Galveston TX) and National Institutes of Health grants RO1 NSO79541-01 to PSS and TKH and R01 EY026089-01A1 to PSS; 2R01 NS073976 grant to TKH, R01-NS100529 to LME; R01s AG033082 and R01 NS065874 to ARL, R01s NS089076 and NS090390 to LMT and Hereditary Disease Foundation Fellowship to CG. We thank Dr. Jean-Paul Vonsattel at the New York Brain Bank, Columbia

University for providing HD patient brain samples. We sincerely thank Drs. Harry Orr and David Konkel for their critical comments. The manuscript was edited by Dr. Lindsay Reese.

## Additional information

### Funding

| Funder | Grant reference number | Author |
|---|---|---|
| NIH Office of the Director | NSO79541-01 | Tapas K Hazra<br>Partha S Sarkar |
| Hereditary Disease Foundation | Postdoctoral Fellowship | Charlene Geater |
| Mitchel Center for Neurodegenerative Diseases, University of Texas Medical Branch | Intramural Developmental Grant | Partha S Sarkar |
| NIH Office of the Director | EY026089-01A1 | Partha S Sarkar |
| NIH Office of the Director | NS100529 | Lisa M Ellerby |
| NIH Office of the Director | AG033082 | Albert R La Spada |
| NIH Office of the Director | NS065874 | Albert R La Spada |
| NIH Office of the Director | NS089076 | Leslie M Thompson |
| NIH Office of the Director | NS090390 | Leslie M Thompson |
| NIH Office of the Director | NS073976 | Tapas K Hazra |

The funders had no role in study design, data collection and interpretation, or the decision to submit the work for publication.

### Author contributions

Rui Gao, Data curation, Formal analysis, Validation, Investigation; Anirban Chakraborty, Data curation, Formal analysis, Validation, Investigation, Methodology, Writing—review and editing; Charlene Geater, Resources, Funding acquisition, Investigation, Methodology; Subrata Pradhan, Formal analysis, Validation, Investigation, Writing—review and editing; Kara L Gordon, Investigation, Methodology; Jeffrey Snowden, Formal analysis, Validation, Investigation, Methodology, Writing—review and editing; Subo Yuan, Data curation, Investigation, Methodology; Audrey S Dickey, Resources, Investigation, Methodology; Sanjeev Choudhary, Resources, Data curation, Formal analysis, Validation, Investigation, Methodology, Writing—review and editing; Tetsuo Ashizawa, Conceptualization, Resources, Methodology, Writing—original draft, Writing—review and editing; Lisa M Ellerby, Resources, Funding acquisition, Methodology, Writing—original draft; Albert R La Spada, Conceptualization, Resources, Supervision, Funding acquisition, Methodology, Writing—review and editing; Leslie M Thompson, Conceptualization, Resources, Supervision, Funding acquisition, Investigation, Methodology, Writing—original draft, Writing—review and editing; Tapas K Hazra, Conceptualization, Resources, Formal analysis, Supervision, Funding acquisition, Methodology; Partha S Sarkar, Conceptualization, Resources, Supervision, Funding acquisition, Visualization, Methodology, Writing—original draft, Project administration, Writing—review and editing

### Author ORCIDs

Sanjeev Choudhary (iD) https://orcid.org/0000-0001-8290-7420
Albert R La Spada (iD) http://orcid.org/0000-0001-6151-2964
Partha S Sarkar (iD) https://orcid.org/0000-0002-2885-8100

### Ethics

Animal experimentation: All procedures involving animals were in accordance with the National Institutes of Health Guide for the care and use of Laboratory Animals, and approved by the Institutional Animal Care and Use Committee of University of California Irivine (protocol #: AUP-18-155); and Duke University (protocol #: A225-17-09).

Decision letter and Author response
Decision letter https://doi.org/10.7554/eLife.42988.032
Author response https://doi.org/10.7554/eLife.42988.033

## Additional files

### Supplementary files
• Transparent reporting form
DOI: https://doi.org/10.7554/eLife.42988.030

### Data availability
All data generated are included in the manuscript and supporting files.

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
