## [Decision Letter]

Thank you for submitting your article "Mutant huntingtin impairs PNKP and ATXN3, disrupting DNA repair and transcription" for consideration by *eLife*. Your article has been reviewed by three peer reviewers, including Harry T Orr as the Reviewing Editor and Reviewer #1, and the evaluation has been overseen by Huda Zoghbi as the Senior Editor. The following individual involved in review of your submission has agreed to reveal their identity: Hayley McLoughlin (Reviewer #2).

The reviewers discussed the reviews with one another and the Reviewing Editor has drafted this decision to help you prepare a revised submission, which should require no additional experiments.

This work investigates a novel transcription-coupled DNA repair complex that includes PNKP, ATXN3 and HTT across many models and through multiple approaches. In the presence of expanded Htt, altered PNKP activity in cells and tissues expressing mutant huntingtin and evidence that transcriptionally active genes contain more strand breaks in HD mouse models were found. These observations suggest hypotheses that are worthy of continued study. However, there are currently insufficient data to support the conclusions of the study and the model put forward by the authors. The conclusions and tone need to be scaled back considerably, including, importantly, statements made in the Abstract and in the Discussion.

Essential revisions:

In the revised manuscript it is essential that conclusions be dampened considerably. Rather than supporting a model based on the existence of an Htt, ATXN3, PNKP complex having a role in transcription as well as DNA repair, reviewers feel strongly the data, as it exists supports the exciting hypotheses that form the basis of important continued study. In addition, it is critical that data be described in a more rigorous manner such that they can be properly interpreted, including a discussion of data that may not align with the principle hypothesis being put forward.

Specific points to essential revisions:

Figure 1:

- There are no QC data to show the quality/purity of the nuclear extracts used in the IPs. This needs to be documented. In particular this brings up the association with HAP-1 that is not typically characterized as a nuclear protein.

- The western blot strips are hard to interpret without any other information, e.g. position of MW markers and in isolation of the rest of the gel(s). It would be helpful to include more extensive western blot data gels as part of supplementary information.

- There appear to be two ATXN3 bands in panel E (also more weakly in F?) showing different intensities in input and IP. Is this real?

- Much of the PLA signal is extranuclear? How does this reconcile with the co-IPs in the nuclear extracts? Notably, the PNKP/ATXN3 PLA signal looks very different here compared to that shown in the Chatterjee 2014 paper where the fluorescent puncta are nuclear with no cytoplasmic signal. In the present manuscript the signal appears mostly cytoplasmic.

- Qualitatively, the nature of the PLA signal is different depending on the pairs. E.g. HTT and ATXN3 compared to HAP1 and ATXN3. This might suggest that these proteins are part of different complexes, depending on their subcellular localization.

More detail/QC needed in Materials and methods and/or figure legends in order to interpret results:

- Overall, the figure legends lack sufficient detail to understand exactly what is being shown. Without detailing every figure, general comments are that the nature of the replicate experiments is not clear. i.e. whether they are biological or technical/how many replicates are performed/what the error (SD or SE) in the data actually represents. Pooling of tissues for animal experiments needs to be clarified. e.g. For the data in Figure 5D-J, where it is stated that 4-5 animals are pooled.. do the error bars then represent technical replicates? Are 4-5 animals pooled for the western blot in panel F? Are G and H data from mice or from human neurons?

- While N-terminal and full-length HTT constructs are specified most of the time, in some cases the "NT" or "FL" designation is omitted and it is not clear what experiment was performed.

- PLA: Figure 1—figure supplement 7 shows a lot of red signal other than that indicated by arrows as "PLA signals", including what appears to be extracellular. It is unclear if this signal is from interacting proteins or is non-specific background. Additional QC in the supplementary figure would be important, as would including the antibody concentrations used.

- PHKP phosphatase activity assay: This needs to be described with some level of detail, not just referring to the authors' previous paper using this method (in which the reader is then referred back to other papers). At least the basic principle of the assay and an explanation of why the phosphatase activity can be specificity ascribed to PHKP in the nuclear extracts.

- Figure 5—figure supplement 2A, B, C: what are the relative levels of mutant and WT HTT expression in the nuclear extracts that provide the phosphatase activity?

- Neuronal differentiation can be very variable. Additional documentation of the characterization of the human neurons is needed. e.g.% neuron vs. glia,% MSNs.

- Rescue data in Figure 5—figure supplement 5: The data showing rescue of altered ATM signaling/DDR with PNKP overexpression are unclear as presented. Figure 5—figure supplement 5F is labeled "mHTT". Is this full-length or N-terminal fragment? Same construct as in Panel D? It is impossible to compare directly the data in panels D and F – the DOX time-course with and without PNKP needs to be carried out in the same experiment to compare protein levels between the two conditions. This is a single western, without quantification and as the data stand it is unclear that the aberrant ATM signaling is being rescued.

Description of HD postmortem brains/Figure 1—figure supplement 7/Figure 2:

- More details are needed: where were the brains obtained from? What are CAG lengths/ages of the HD brains and ages of the control brains? What is the pathological grade (if known)? Ideally the CAG length(s), as genotyped, should be reported. N=4 HD brains are reported in the Materials and methods, but the data on these 4 brains are unclear – only two appear to be specifically mentioned in the Results.

- In Figure 1—figure supplement 7 and Figure 2- "brain sections" is very vague. What are the actual brain regions analyzed?

- Which brains are shown in Figure 1—figure supplement 7?

- The brains shown in Figure 2 are designated as Q94 and Q82. It should be pointed out by the authors that these repeat lengths are associated with juvenile-onset HD. Were any adult onset cases examined?

- It is not accurate to report the colocalization data in the brains (Figure 2) as test or evidence of interaction.

Huntingtin-PNKP interaction data:

- Data in support of increased interaction of PNKP PHOS+KIN domain with Myc-NT-mHTT-Q97 relative to Myc-NT-HTT-Q23 (Figure 4C+D): The quality of the western blot in Figure 4C is very poor. Essentially, this is the only piece of data supporting an increased interaction of any form of mutant huntingtin with PNKP, and the data need to be much clearer to support this. Increased binding of the mutant N-terminal fragment to PNKP PHOS+KIN is not obviously supported by the BiFC data (Figure 4E). There does not seem to be an altered interaction of the N-terminal fragments with FL-PNKP (Figure 4C+D); what does this mean for the complex in vivo that presumably contains full length PNKP? When full length HTT is overexpressed (Figure 3 D and E), there also does not seem to be a stronger interaction either of PHOS+KIN of FL-PNKP with Myc-FL-HTT-Q148 relative to Myc-FL-HTT-Q23. This is not commented on/taken into account in the interpretation of the data. Is there any evidence for differences in interaction between mutant and WT HTT with PNKP outside of N-terminal fragment overexpression?

- Figure 4A shows relatively more co-IP of PNKP in GFP-NT-mHTT-Q97 expressing cells relative to GFP-NT-mHTT-Q23 (Figure 4A). This effect is not apparent in a comparison of the IPs from Myc-NT-HTT-Q23 and Myc-NT-mNTT-Q148 (Figure 4B). These observations are not discussed and their significance and the apparent difference between them are not clear.

---

## [Author Response]

Essential revisions:In the revised manuscript it is essential that conclusions be dampened considerably. Rather than supporting a model based on the existence of an Htt, ATXN3, PNKP complex having a role in transcription as well as DNA repair, reviewers feel strongly the data, as it exists supports the exciting hypotheses that form the basis of important continued study. In addition, it is critical that data be described in a more rigorous manner such that they can be properly interpreted, including a discussion of data that may not align with the principle hypothesis being put forward.Specific points to essential revisions:Figure 1:- There are no QC data to show the quality/purity of the nuclear extracts used in the IPs. This needs to be documented. In particular this brings up the association with HAP-1 that is not typically characterized as a nuclear protein.

The purity of the nuclear and cytosolic protein fractions was assessed by immunoblotting for GAPDH and hnRNPC1/C2 as cytosolic and nuclear protein markers respectively. Absence of hnRNPC1/C2 protein in cytosolic fraction and GAPDH in nuclear fraction clearly demonstrate that these fractions are enriched and free of any cross-contamination (Figure 1A in the revised version). Our data shows the presence of HTT, ATXN3, PNKP and HAP-1 in the nuclear as well as cytosolic protein fractions.

- The western blot strips are hard to interpret without any other information, e.g. position of MW markers and in isolation of the rest of the gel(s). It would be helpful to include more extensive western blot data gels as part of supplementary information.

Figure 1 of the revised manuscript is modified to indicate the positions of Molecular Weight markers for each protein. The original western blots for the processed immuno-precipitation (IP) experiments shown in Figure 1 are now compiled in Figure 1—figure supplements 1-6. Different sections of the blotting membrane was probed with variety of antibodies to analyze the components of the molecular protein complex. Electrophoretic gels from at least two independent IP experiments were run in duplicates to check reproducibility of these results. As per reviewer’s suggestions, figures generated from duplicate WB and the corresponding unprocessed blots were compiled in Figure 1—figure supplements 1-6.

- There appear to be two ATXN3 bands in panel E (also more weakly in F?) showing different intensities in input and IP. Is this real?

We apologize for the poor quality of the IP/WBs in the mentioned figure, particularly the ATXN3 WB. These IP experiments were repeated under enhanced stringent conditions (IP buffer containing 200 mM KCl) and the improved IP/WB images from these repeated experiments are now added to the revised manuscript (Figures 1F, 1G and Figure 1—figure supplements 5 and 6). These new figures should remove any ambiguity in data interpretation.

- Much of the PLA signal is extranuclear? How does this reconcile with the co-IPs in the nuclear extracts? Notably, the PNKP/ATXN3 PLA signal looks very different here compared to that shown in the Chatterjee 2014 paper where the fluorescent puncta are nuclear with no cytoplasmic signal. In the present manuscript the signal appears mostly cytoplasmic.

We agree with the reviewer that the PLA signal as reported in the Figure 1 is present in the nuclei, nuclear periphery, and the cytoplasm. New unpublished data from our laboratory suggest that both WT and mutant HTT are also present in mitochondria and forms mitochondrial complex with ATXN3 and PNKP similar to the nuclear complex of HTT with these proteins. Our data show that HTT, PNKP, and ATXN3 form a multi-protein complex with mitochondrial RNA polymerase (POLRMT) and mitochondrial transcription factors. We hypothesize that this complex probably performs mtDNA repair during mtDNA transcription. These results will be a part of our next manuscript.

Additionally, the modified protocol used in the current manuscript is more stringent as compared to that of Chatterjee 2015. The manuscript is revised appropriately to clearly address this concern.

- Qualitatively, the nature of the PLA signal is different depending on the pairs. E.g. HTT and ATXN3 compared to HAP1 and ATXN3. This might suggest that these proteins are part of different complexes, depending on their subcellular localization.

The intensity and relative abundance of the PLA signal could vary due to several factors, such as abundance, size, conformation and the antibody-recognition epitope of each protein as well as the quality of the antibody used. Although the possibility of these proteins being part of different complexes cannot be completely ruled out, the stringent condition (200-400 mM KCl) used during IP and reverse IPs (data shown in Figure 1, and Figure 1—figure supplements 1-6) markedly reduces this possibility. Nevertheless, we do agree that additional structural studies using purified proteins would be necessary to understand these interactions in multi-protein complex. These studies will be part of future communication from our laboratory.

More detail/QC needed in Materials and methods and/or figure legends in order to interpret results:

As per reviewer’s suggestion, the current manuscript is suitably revised to provide detailed information about each figure and Experimental Procedures.

- Overall, the figure legends lack sufficient detail to understand exactly what is being shown. Without detailing every figure, general comments are that the nature of the replicate experiments is not clear. i.e. whether they are biological or technical/how many replicates are performed/what the error (SD or SE) in the data actually represents. Pooling of tissues for animal experiments needs to be clarified. e.g. For the data in Figure, 5D-J, where it is stated that 4-5 animals are pooled.. do the error bars then represent technical replicates? Are 4-5 animals pooled for the western blot in panel F?

We sincerely apologize for the brief figure legends in the original manuscript that could be confusing. We have now provided detailed information regarding animal usage and additional required specifics. The PNKP activity and total repair assays were performed in transgenic and age-matched WT mice. 15 age-matched animals were used in each group. Tissues from CTX, STR or CRBL dissected from a group of 4 to 5 transgenic or control brain were pooled together to purify nuclear fraction for various analysis shown in Figure 5. Every experiment was repeated minimum of three times. Pooled samples were also used in panel 5F. The figure legends in the manuscript is revised to reflect all the details of animal usage, replications, and statistical analysis specific to each figure.

Are G and H data from mice or from human neurons?

This data is from induced pluripotent stem cell (iPSC)-derived human primary neurons. We have changed the figure legends to clarify this confusion.

- While N-terminal and full-length HTT constructs are specified most of the time, in some cases the "NT" or "FL" designation is omitted and it is not clear what experiment was performed.

In the revised manuscript, we have included FL or NT to designate the full-length and N-terminal of HTT, respectively in relevant places.

- PLA: Figure 1—figure supplement 7 shows a lot of red signal other than that indicated by arrows as "PLA signals", including what appears to be extracellular. It is unclear if this signal is from interacting proteins or is non-specific background. Additional QC in the supplementary figure would be important, as would including the antibody concentrations used.

We agree that the PLA signal is present in the nuclei, nuclear periphery, and the cytoplasm. Recent unpublished data from our laboratory show that the cytoplasmic signals are actually from a similar complex that is detected in the mitochondria. We find that similar HTT-PNKP complex is also present in the mitochondria, and associates with mitochondrial DNA. We postulate that this mitochondrial complex could potentially be involved in mtDNA repair similar to its nuclear counterpart. We have included this explanation in the revised manuscript. The molecular mechanism explaining the role of said complex in mtDNA repair will be a part of our next manuscript.

- PHKP phosphatase activity assay: This needs to be described with some level of detail, not just referring to the authors' previous paper using this method (in which the reader is then referred back to other papers). At least the basic principle of the assay and an explanation of why the phosphatase activity can be specificity ascribed to PHKP in the nuclear extracts.

We have provided a detailed protocol and an appropriate reference for the PNKP phosphatase assay in the Materials and methods section of the revised version.

- Figure 5—figure supplement 2A, B, C: what are the relative levels of mutant and WT HTT expression in the nuclear extracts that provide the phosphatase activity?

We have now added WBs to show the expression levels of mutant and WT HTT expressions in these cells in Figure 5—figure supplement 2C.

- Neuronal differentiation can be very variable. Additional documentation of the characterization of the human neurons is needed. e.g.% neuron vs. glia,% MSNs.

We have characterized the primary neurons and provided the approximate percentage of neurons and glia in the cell populations. We see >98% neuronal populations and about 2% glial cells in our culture.

- Rescue data in Figure 5—figure supplement 5: The data showing rescue of altered ATM signaling/DDR with PNKP overexpression are unclear as presented.

We have provided an improved figure (Figure 5—figure supplement 5) to show DDR-ATM pathway in mutant cells and in mutant cells over-expressing exogenous PNKP. We have shown phosphorylation-dependent activation of two ATM target proteins e.g., p53 (phosphorylation of p53 at serine 15) and histone H2A (S139) in the revised manuscript. Quantification of the WBs show that mHTT-mediated activation of the DDR pathway is significantly inhibited in the cells over-expressing PNKP.

Figure 5—figure supplement 5F is labeled "mHTT". Is this full-length or N-terminal fragment?

The N-terminal fragment of mutant HTT-Q97 were used in the experiments described in Figure 5—figure supplement 5A to F.

PC12 cells expressing the full-length wt-HTT-Q23 (FL-wtHTT-Q23) and mHTT-Q148 (FL-mHTT-Q148) were used for Caspase-3 and cell toxicity measurements by FACS, as shown in Figure 5—figure supplement 5G, H and I. We have made the appropriate changes in the figure and the corresponding figure legends.

Same construct as in Panel D? It is impossible to compare directly the data in panels D and F – the DOX time-course with and without PNKP needs to be carried out in the same experiment to compare protein levels between the two conditions. This is a single western, without quantification and as the data stand it is unclear that the aberrant ATM signaling is being rescued.

We appreciate reviewers to bring this to our attention. We have provided quantification of the WBs in Figure 5—figure supplement 5D and E demonstrating the significant inhibition of mHTT-mediated DDR pathway activation in the PNKP over-expressing cells.

Description of HD postmortem brains/Figure 1—figure supplement 7/Figure 2:- More details are needed: where were the brains obtained from? What are CAG lengths/ages of the HD brains and ages of the control brains? What is the pathological grade (if known)? Ideally the CAG length(s), as genotyped, should be reported. N=4 HD brains are reported in the Materials and methods, but the data on these 4 brains are unclear – only two appear to be specifically mentioned in the Results.- In Figure 1—figure supplement 7 and Figure 2- "brain sections" is very vague. What are the actual brain regions analyzed?

The brain sections in Figure 2 are from the Caudate-Putamen, and Accumbens (CPA). Tissues were obtained from early onset HD patients (mutant HTT encoding Q82 and Q84; grade 4/4, and manifesting severe phenotypes) and age-matched control brains. This additional information has been added to the Materials and methods section and the respective figure legends in the revised manuscript.

- Which brains are shown in Figure 1—figure supplement 7?

In Figure 1—figure supplement 7, we have shown data using brain sections (Caudate-Putamen-Accumbens) from a 42 years old HD patient (mutant HTT encoding Q55, disease grade 4/4, and manifesting severe phenotypes) and age-matched normal brain tissue. This additional information is now included in the figure legend.

- The brains shown in Figure 2 are designated as Q94 and Q82. It should be pointed out by the authors that these repeat lengths are associated with juvenile-onset HD. Were any adult onset cases examined?

Thank you for this suggestion. The figure legends in the revised manuscript clearly display these details. We have also examined HD brain sections from 42 and 49-year-old adult onset HD patients (mutant huntingtin encoding 55Qs, 58Qs) and observed similar co-localization of PNKP and ATXN3 with HTT.

- It is not accurate to report the colocalization data in the brains (Figure 2) as test or evidence of interaction.

We appreciate reviewer’s suggestion. Necessary changes been made in the text.

Huntingtin-PNKP interaction data:- Data in support of increased interaction of PNKP PHOS+KIN domain with Myc-NT-mHTT-Q97 relative to Myc-NT-HTT-Q23 (Figure 4C+D): The quality of the western blot in Figure 4C is very poor. Essentially, this is the only piece of data supporting an increased interaction of any form of mutant huntingtin with PNKP, and the data need to be much clearer to support this.

We have repeated this IP/WB experiment to improve the quality of Figure 4C. A much-improved image is now included in the revised manuscript that clearly shows the interaction between NT-wtHTT-Q23, FL-PNKP and PNKP-(PHOS+KIN). The interaction strength of PNKP-(PHOS+KIN) domain with WT and mHTT cannot be reliably quantified via the Co-IP experiment, which was conducted with transient overexpression of the proteins. We have substantially revised the conclusive statement made regarding the increased interaction of PNKP-(PHOS+KIN) with NT-mHTT-Q97. Further structural investigations will be required to determine the effects of polyQ expansion on the HTT and PNKP-(PHOS+KIN) interaction.

Increased binding of the mutant N-terminal fragment to PNKP PHOS+KIN is not obviously supported by the BiFC data (Figure 4E). There does not seem to be an altered interaction of the N-terminal fragments with FL-PNKP (Figure 4C+D); what does this mean for the complex in vivo that presumably contains full length PNKP? When full length HTT is overexpressed (Figure 3D and E), there also does not seem to be a stronger interaction either of PHOS+KIN of FL-PNKP with Myc-FL-HTT-Q148 relative to Myc-FL-HTT-Q23. This is not commented on/taken into account in the interpretation of the data. Is there any evidence for differences in interaction between mutant and WT HTT with PNKP outside of N-terminal fragment overexpression?

We thank the reviewers for pointing out this important issue regarding the relative binding affinity between FL-PNKP and WT or mutant HTT. We agree that the binding affinities of PNKP-(PHOS+KIN) domain with WT and mHTT cannot be assessed accurately by the Co-IP experiment which is based on transient overexpression of relevant proteins. Several factors may modulate the outcome e.g., transfection efficiency, appropriate expressions, or folding of the fused peptides. More rigorous biophysical measurements using purified proteins/peptides would be required to understand the true nature of these protein-protein interactions. Since HTT-TCR complex is not completely characterized, there may be additional unidentified factors in the complex that could significantly alter these interactions in vivo. Alternatively, the binding of the mHTT with FL-PNKP or PNKP-(PHOS+KIN) domain may not be different but the expanded polyQ sequences in mHTT may physically hinder the catalytic site of PNKP thus depleting its activity and DNA repair. These possibilities have now been added in the revised version.

- Figure 4A shows relatively more co-IP of PNKP in GFP-NT-mHTT-Q97 expressing cells relative to GFP-NT-mHTT-Q23 (Figure 4A). This effect is not apparent in a comparison of the IPs from Myc-NT-HTT-Q23 and Myc-NT-mNTT-Q148 (Figure 4B). These observations are not discussed and their significance and the apparent difference between them are not clear.

These co-IP experiments were done to assess cellular interactions of N-terminal fragments of HTT with PNKP. The IP/WB data described in Figure 4A using GFP-NT-wtHTT and GFP-NT-mHTT suggests that while both wtHTT and mHTT interact with endogenous PNKP and the associated TCR components, interaction with the mutant was noticeably stronger. We agree with the reviewer’s comment that no distinction was observed between mutant and wild type HTT interaction with PNKP or PNKP domains in the co-IP experiments performed with overexpressing Myc-NT-mHTT-Q148 and Myc-NT-wtHTT-Q23 and various PNKP domains (Figure 4C and D). It is possible that the addition of MYC and GFP tag on these peptides may alter the conformation of the peptides distinctly and that may in turn influence their interaction efficacy with other proteins. Our data is a representation of qualitative interaction between these proteins and should not be interpreted as quantitative measurement of these interactions. To make quantitative assessments and to characterize the nature of these protein-protein interactions, structural and biophysical studies using purified proteins would be necessary. Currently, these experiments are being conducted in our laboratory and will be a part of future publications. We have discussed these possibilities in the revised manuscript.